# The deubiquitinase OTUD1 regulates immunoglobulin production and proteasome inhibitor sensitivity in multiple myeloma

Alexander Vdovin [1,2,3], Tomas Jelinek[1,2], David Zihala[1,2,3], Tereza Sevcikova[1,2,3], Michal Durech [1,2], Hana Sahinbegovic[1,2,3], Renata Snaurova [1,2,3], Dhwani Radhakrishnan[1,2,3], Marcello Turi [1,2,3], Zuzana Chyra[1,2], Tereza Popkova[1,2], Ondrej Venglar[1,2,3], Matous Hrdinka [1,2], Roman Hajek[1,2] & Michal Simicek [1,2,3] ✉

Serum monoclonal immunoglobulin (Ig) is the main diagnostic factor for patients with multiple myeloma (MM), however its prognostic potential remains unclear. On a large MM patient cohort (n = 4146), we observe no correlation between serum Ig levels and patient survival, while amount of intracellular Ig has a strong predictive effect. Focused CRISPR screen, transcriptional and proteomic analysis identify deubiquitinase OTUD1 as a critical mediator of Ig synthesis, proteasome inhibitor sensitivity and tumor burden in MM. Mechanistically, OTUD1 deubiquitinates peroxiredoxin 4 (PRDX4), protecting it from endoplasmic reticulum (ER)-associated degradation. In turn, PRDX4 facilitates Ig production which coincides with the accumulation of unfolded proteins and higher ER stress. The elevated load on proteasome ultimately potentiates myeloma response to proteasome inhibitors providing a window for a rational therapy. Collectively, our findings support the significance of the Ig production machinery as a biomarker and target in the combinatory treatment of MM patients.

Multiple myeloma (MM) is a malignancy of immunoglobulin (Ig)-producing plasma cells and the second most common hematological cancer[1]. In the last decade, the life expectancy of MM patients has improved significantly, mainly due to the introduction of novel treatment options, with proteasome inhibitors (PIs) being the outstanding pioneers of rational anti-MM therapy[2]. Three representatives of PIs (bortezomib, carfilzomib, ixazomib) have been approved and are commonly used in routine clinical practice. Even though PIs presented a tremendous success in MM therapy, they are currently not included as a backbone in all anti-myeloma regimens. In fact, PIs-free triplet - daratumumab, lenalidomide and dexamethasone - has become a new standard of care in newly diagnosed, transplant-ineligible MM patients with unprecedented outcomes. Indeed, the majority of newly diagnosed myeloma patients will not receive PIs in their first line therapy, thus a reliable biomarker that would predict the favorable response to PIs is eagerly needed[3].

It is generally accepted that the tremendous capacity to produce Ig molecules is a prerequisite for the unique myeloma sensitivity to PIs[4]. The secreted serum monoclonal Ig (M-protein) is considered as a diagnostic hallmark of all monoclonal gammopathies and the kinetic of

[1]Department of Hematooncology, Faculty of Medicine, University of Ostrava, OstravaSyllabova 19, 703 00, Czech Republic. [2]Department of Hematooncology, University Hospital Ostrava, Ostrava17. listopadu 1790/5, 708 00, Czech Republic. [3]Faculty of Science, University of Ostrava, Ostrava30. dubna 22, 701 03, Czech Republic. ✉e-mail: michal.simicek@fno.cz

serum M-protein levels is used to monitor the disease[5]. Classical hematological response criteria to the treatment are based on the decrease and/or disappearance of M-protein[6]. A sudden rise in the serum M-protein is a characteristic feature of the upcoming relapse and is considered to reflect a higher tumor burden[7,8]. To fully engage the clinical potential of this exclusive myeloma attribute, a deeper understanding of the molecular mechanisms behind Ig production is essential.

During the process of plasma cell maturation, both transcription and synthesis of Ig markedly increase[9]. This puts enormous pressure on endoplasmic reticulum (ER) folding machinery to generate correctly assembled Ig molecules. Overloading the ER folding capacity ultimately leads to the accumulation of misfolded proteins that need to be extracted from ER, ubiquitinated, and targeted for degradation in proteasome[10–13]. An increase in ubiquitinated species creates an imbalance in proteasome load versus capacity in both normal and malignant plasma cells, making them exceptionally prone to apoptosis upon proteasome inhibition[14–17].

Tight control of protein homeostasis is therefore critical for myeloma cells' survival and any disruption of the proteosynthetic and proteolytic machinery is deleterious. This was proven by many mechanistic studies associating PI resistance with mutations and expression changes of proteasome and ribosome subunits, as well as the ER stress components[18–20]. The prognostic potential of these parameters in the clinical setting is, however, not entirely accepted. Therefore, there is a constant need for robust assays to identify PI-insensitive MM patients who could profit from precision combination therapy that would eradicate the resistant clones in the early stage of the disease.

Here, we identify levels of aberrant plasma cell intracellular Ig as an independent prognostic factor which could distinguish MM patients suitable for successful PI-based treatment. Additionally, we uncover regulatory mechanism driving Ig synthesis at the translation level.

## Results

### Intracellular Ig and OTUD1 but not serum M-protein predict outcome of MM patients

Tremendous proteosynthetic capacity of plasma cells is utilized in the treatment of MM patients with PIs[2]. In addition to sensitivity to PIs, extreme amount of newly synthetized Ig molecules creates a burden of constantly elevated ER stress, high energy demands and nutrients consumption[9]. Therefore, Ig synthesis can be one of the parameters of clonal selection in the development and progression of MM. Aberrant plasma cell clones which lost the ability to synthesize complete Ig molecule and produce only Ig light chain (IgL) form usually more aggressive tumors[21–23]. However, those are relatively rare cases. Up to date, it has not been studied if variability in Ig production affects disease outcome in patients with secretory MM.

A common diagnostic parameter measured in all patients with plasma cell dyscrasias is the level of serum M-protein (secreted monoclonal Ig). To thoroughly validate M-protein prognostic value, we analyzed progression-free survival (PFS) and overall survival (OS) in up to date the largest cohort of newly diagnosed MM patients (Supplementary Fig. 1a, b and Supplementary Table 1) and further, in a subgroup of patients who received bortezomib in the first line of therapy (Supplementary Fig. 1c, d and Supplementary Table 2). We did not observe any correlation between the levels of serum M-protein at the time of diagnosis and either PFS or OS, suggesting that amount of secreted M-protein does not reflect tumor aggressiveness or drug sensitivity. On the other hand, concentration of intracellular Ig (iIg) should not be affected by tumor size and thus, it could better reflect the proteosynthetic rate and response of MM patients to PI-based therapy.

In this study we measured amount of iIgL in the aberrant plasma cells from a cohort of 86 newly diagnosed MM patients. We observed a positive correlation between iIgL and PFS (Fig. 1a; Supplementary Table 3). MM patients with iIgL concentration below 0.1 µg iIgL/µg of total plasma cell proteins (iIgL low) had a worse prognosis compared

to patients with iIgL above 0.1 µg iIgL/µg of total plasma cell proteins (iIgL high) (Fig. 1b). The cubic spline analysis underlined the validity of the selected cut-off value for iIgL concentration (Fig. 1c). Multivariate analysis further supported significance of iIgL prognostic effect over other commonly used predicting factors (Supplementary Table 4). Additionally, we analyzed relative iIgL levels in 106 newly diagnosed MM patients by flow cytometry. This independent approach also confirmed the prognostic impact of iIgL (Supplementary Fig. 1e).

It is well established that the production of Ig is regulated on a transcriptional level by plasma cell-specific factors including IRF4 and XBP1[18,24]. However, the expression analysis of primary MM patient samples revealed that the differences in the iIgL protein content could not be explained by altered transcription from the IGL locus (Fig. 1d). Similarly, analysis of the public expression dataset showed no association of IGLC1 expression with the MM patients' survival (Fig. 1e). Together, these data highlight the importance of post-transcriptional regulation of Ig production in regards to disease outcome.

The enormous synthesis of Ig molecules in plasma cells sets high demands on the folding apparatus and is inherently connected with an extensive load of misfolded, highly ubiquitinated proteins[4]. Therefore, we hypothesized that deubiquitinating enzymes (DUBs) might play a significant role in the Ig production pathway. To identify DUBs controlling Ig synthesis, we performed a CRISPR knockout screen in myeloma cells covering most of the currently annotated human DUBs and looked for those that alter Ig production and at the same time affect MM survival (Supplementary Fig. 1f). From the candidate list (Supplementary Table 5), we selected DUBs that are highly expressed in plasma cells (Supplementary Data 1)[25]. Ovarian Tumor Deubiquitinase 1 (OTUD1) was the only DUB that fulfilled all the applied criteria. Moreover, OTUD1 is the most highly expressed DUB in B cells compared to other hematopoietic cells (Supplementary Fig. 1g) and reaches its expression peak in the fully differentiated bone marrow plasma cells throughout B-cell development (Supplementary Fig. 1h). Altogether, the cumulative evidence indicates a crucial role of OTUD1 in plasma cell biology, particularly in Ig synthesis.

To evaluate the effect of OTUD1 on Ig production in our cohort, we split the patients based on the median OTUD1 expression (OTUD1 low and OTUD1 high) and quantified the iIgL content. The aberrant plasma cells isolated from the OTUD1 low MM patients had less iIgL compared to the OTUD1 high group (Fig. 1f). Similarly, dividing the patients according to the median iIgL concentration (0.1 µg iIgL/µg of total plasma cell proteins) revealed differences in the amount of OTUD1 mRNA (Fig. 1g). Yet again, the various quantities of iIgL in the OTUD1 low and OTUD1 high MM patients could not be explained by changes in the transcription of the IGL genes (Supplementary Fig. 1i). Importantly, increased expression of OTUD1 correlated with better survival of MM patients both in our cohort (Supplementary Fig. 1j) and in publicly available datasets (Fig. 1h, i). Collectively, these results suggest that OTUD1 regulates production of IgL on the post-transcriptional level and OTUD1 expression has a similar prognostic potential as iIgL concentration in newly diagnosed MM patients.

### OTUD1 is a regulator of Ig production and proliferation in myeloma cells

To further study the involvement of OTUD1 in Ig production, we generated a panel of MM cell lines with doxycycline (dox)-inducible OTUD1 overexpression (OTUD1 oe) and shRNA-mediated OTUD1 knock-down (sh OTUD1_1, sh OTUD1_2) (Fig. 2a–c). As controls, we used isogenic cells without dox induction (control) for OTUD1 oe and cells expressing non-mammalian shRNA (sh control) for sh OTUD1 cells. Similar to MM patients, we observed a significant increase of iIgL in cells with OTUD1 oe (Fig. 2d). On the contrary, iIgL dropped in cells expressing sh OTUD1 (Fig. 2e). In both cases, OTUD1 did not influence the expression of IgL mRNA (Supplementary Fig. 2a, b) nor the amount of secreted IgL (Supplementary Fig. 2c). Further, the knock-down of other DUBs did

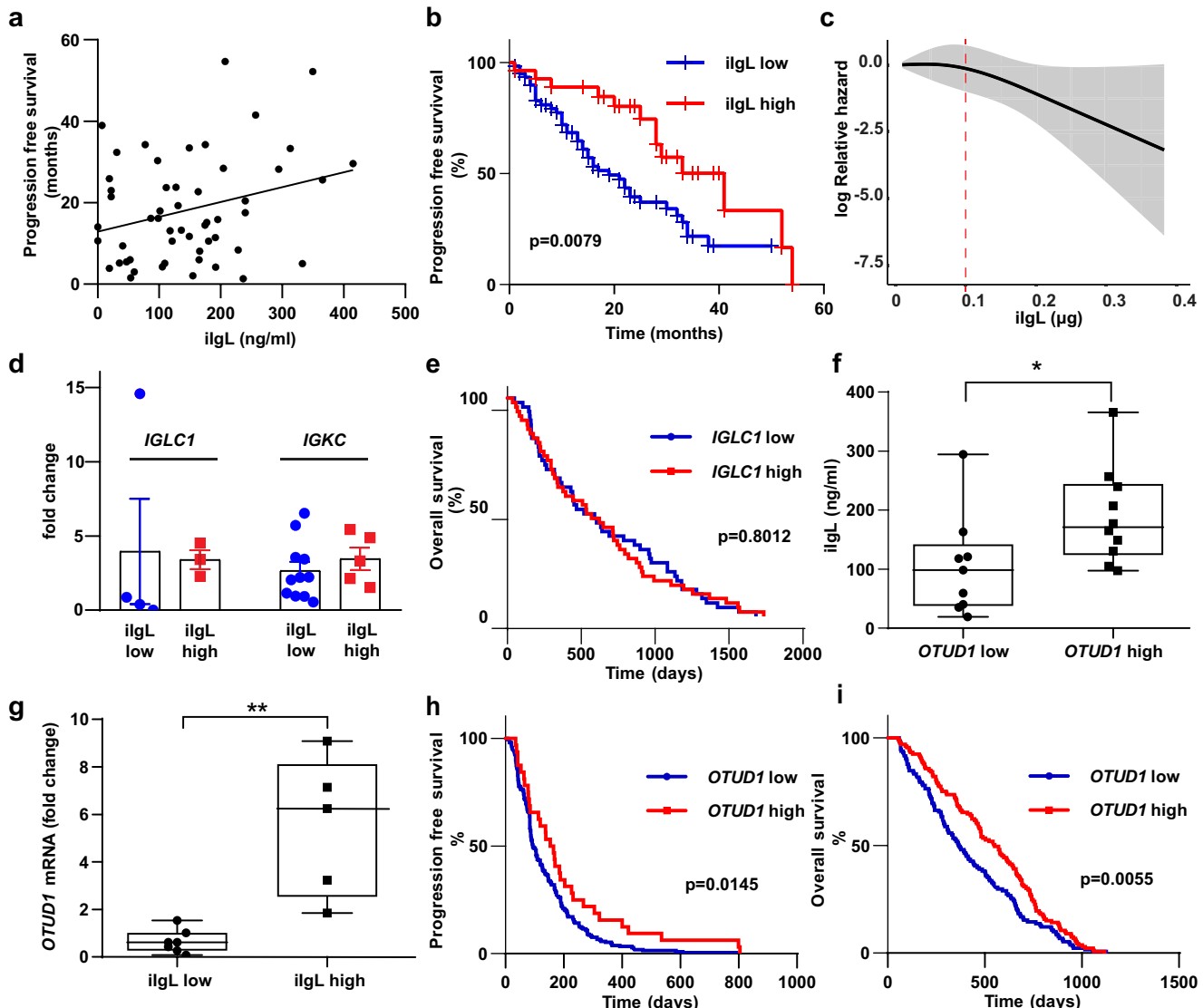

**Fig. 1 | iIg and OTUD1 are predictors of myeloma survival. a** Correlation analysis of iIgL and PFS of the newly diagnosed MM patients ($n = 53$). iIgL was quantified using ELISA in lysates from aberrant plasma cells sorted using the myeloma-specific panel. Significance of regression coefficient was determined by $F$-test (DF$n = 1$, DFd = 51, $p = 0.0387$; R square = 0.08112, 95% Confidence Intervals: slope 3.921 to 141.6, Y-intercept 6.807 to 18.99, X-intercept −4.577 to −0.05085) **b** PFS was evaluated in MM patients ($n = 86$) divided into two groups based on the iIgL content. iIgL high (red) and iIgL low (blue) groups were determined by the median iIgL concentration (0.1 μg iIgL/μg total plasma cell proteins). Significance was compared using the log-rank (Mantel-Cox) test ($n = 59$ for iIgL low, $n = 27$ for iIgL high, $p = 0.0079$). **c** Cubic splines analysis (knots = 3) showing log relative hazard of progression as a function of the level of iIgL. Shaded area represents 95% confidence intervals. **d** Expression of *IGLC1* and *IGKC* was analyzed by RT-PCR in the iIgL high and iIgL low groups from (**b**), (*IGLC1*: $n = 5$ for iIgL low, $n = 5$ for iIgL high; *IGKC*: $n = 9$ for iIgL low, $n = 5$ for iIgL high). Significance was compared using the two-

tailed Student's t test. Error bars represent the mean ± SD (for *IGLC1* $p = 0.9003$, for *IGKC* $p = 0.45$). **e** Overall survival of newly diagnosed MM patients' groups based on *IGLC1* expression in aberrant plasma cells ($n = 273$, dataset GSE2658). Significance was compared using log-rank test **f, g** Data obtained from MM patient samples used in (**b**). *OTUD1* expression was analyzed by RT-PCR and samples were divided into the *OTUD1* low and *OTUD1* high groups according to the median *OTUD1* expression (number of biological samples $n = 9$ for *OTUD1* low, $n = 10$ for *OTUD1* high). iIgL content was measured by ELISA and samples were divided into the iIgL low and iIgL high groups ($n = 7$ for iIgL low, $n = 5$ for iIgL high). Significance was compared using the two-tailed Student's $t$ test (in **f** $p = 0.0221$, in **g** $p = 0.0014$). The whiskers represent minimal and maximal values, the box extends from the 25th to 75th percentiles. PFS (**h**) and OS (**i**) of MM patients' groups based on the *OTUD1* expression in aberrant plasma cells ($n = 414$, dataset GSE4581). Significance was compared using a log-rank test (in **h** $p = 0.0145$, in **i** $p = 0.0055$).

not affect IgL production in MM cells (Supplementary Fig. 2d), supporting the specific effect of OTUD1. Finally, the expression of the catalytically inactive OTUD1 C320R mutant did not cause any changes in iIgL levels (Supplementary Fig. 2e), therefore OTUD1 enzymatic activity is required to sustain high IgL production.

Because both our and public data imply a potential inhibitory effect of *OTUD1* expression on myeloma aggressiveness, we tested the proliferation capacity of our OTUD1 genetic models. As expected, OTUD1 oe suppressed myeloma cell growth while introduction of sh OTUD1

promoted proliferation (Fig. 2f, g). In the in vivo settings, the effect of differential OTUD1 expression on tumor growth was even more pronounced (Fig. 2h-k; Supplementary Fig. 2f, g). Additionally, the concentration of iIgL in myeloma cells extracted from the mouse xenografts correlated with OTUD1 levels (Fig. 2l, m). Further analysis revealed that OTUD1 oe induced partial arrest in the S phase of the cell cycle without affecting cell viability (Supplementary Fig. 2h, i). These results recapitulate the situation seen in MM patients and validate the use of cell line models to study the function of OTUD1 in myeloma pathogenesis.

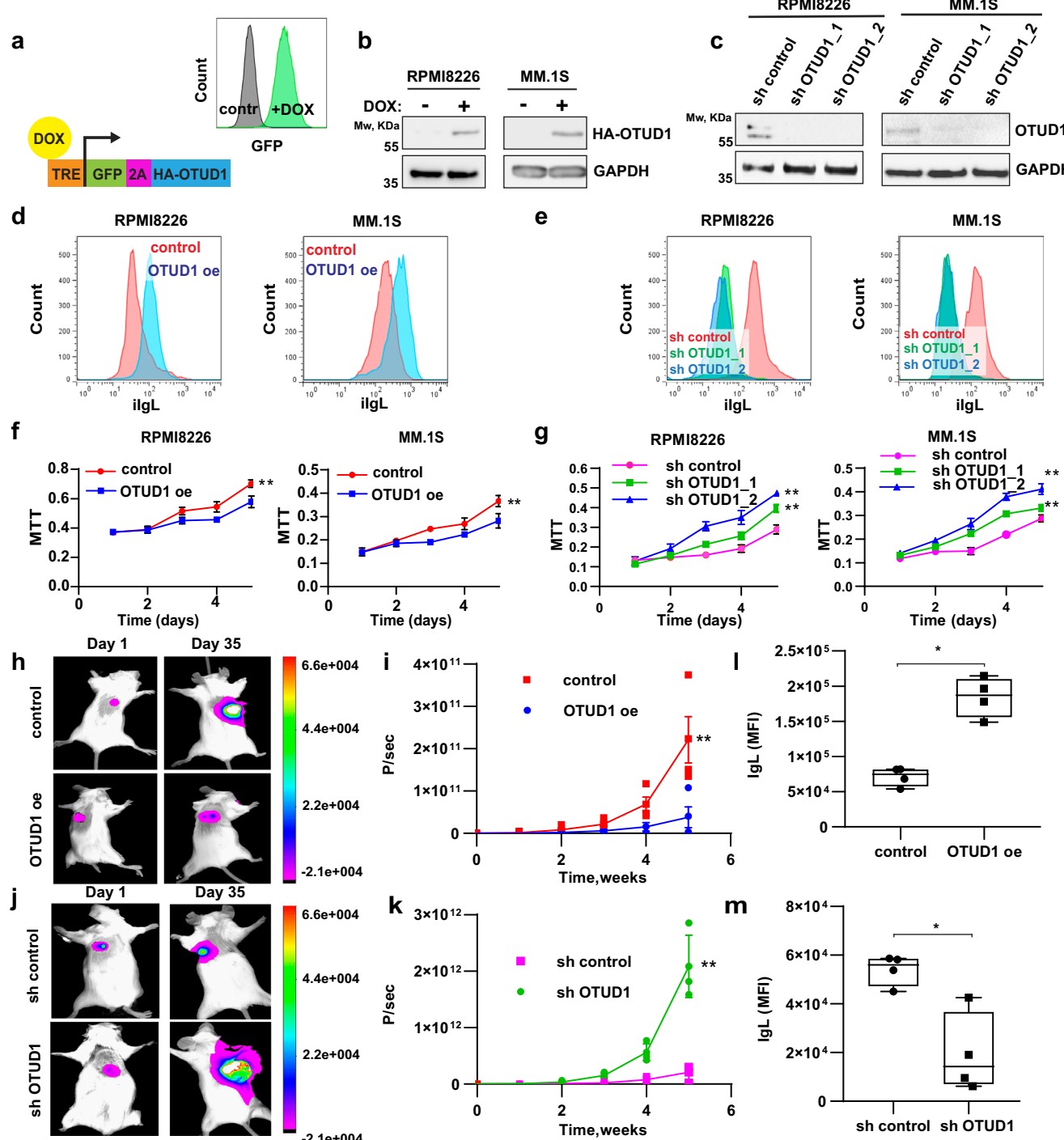

## OTUD1 modulates IgL production, proteasome load and PI sensitivity

While exploring the mechanism behind the modulation of iIgL levels by OTUD1, we noticed dramatic changes in total ubiquitin pools in myeloma cells with altered OTUD1 expression. Surprisingly, OTUD1 oe led to a massive rise in total ubiquitination (Fig. 3a), while OTUD1 knock-down had the opposite effect (Fig. 3b). At the same time, the observed changes in protein ubiquitination directly correlated with the ER stress status (Fig. 3a–d).

As we observed this phenomenon only in myeloma but no other cell lines (Supplementary Fig. 3a), and OTUD1 regulates IgL production in plasma cells (Fig. 2d, e), we hypothesized that the differences in global ubiquitination might be caused by alteration in the iIgL levels.

Indeed, siRNA-mediated IgL knock-down significantly reduced the ubiquitination pattern (Fig. 3e).

Additionally, immunoprecipitation of IgL from myeloma cells with OTUD1 oe revealed an increase in IgL ubiquitination (Fig. 3f) while ubiquitination of IgL dropped in sh OTUD1 cells (Fig. 3g). Because IgL is a secreted protein, it is not exposed to cytosolic ubiquitin ligases. Therefore, we speculated that the ubiquitinated forms of IgL represent the damaged or misfolded molecules retranslocated from ER and destined for proteasome degradation. Indeed, the application of the cell-permeable proteasome fluorescent substrate showed a significant occupation of proteasome in OTUD1 oe cells which was reverted by the introduction of siRNA-targeting IgL (Fig. 3h). Altogether, our results indicate that OTUD1 regulates synthesis of IgL and the vast amount of

**Fig. 2 | OTUD1 drives Ig production and tumor growth in MM. a** Schematic illustration of the construct used for inducible overexpression (oe) of OTUD1. Tet Response Element (TRE) upstream of a promoter activates gene expression after doxycycline (dox) binding. GFP and HA-tagged OTUD1 are regulated by the same promoter and separated by 2A self-cleaving peptide. After 3 days of dox induction expression of GFP can be detected by flow cytometry. **b** Western blot analysis of HA-OTUD1 expression after dox induction in RPMI8226 and MM.1S cells. **c** Western blot analysis of endogenous OTUD1 in MM cell lines RPMI8226 and MM.1S expressing non-mammalian control shRNA (sh control), or two different shRNAs targeting OTUD1 (sh OTUD1_1 and sh OTUD1_2). **d, e** iIgL content was analyzed by intracellular staining and flow cytometry in OTUD1 oe or sh OTUD1 cells and the respective isogenic controls (representative experiment, $n = 3$). $10^4$ of MM cells with OTUD1 oe (**f**) or sh OTUD1 (**g**) were plated for each time point. Cell proliferation was estimated by MTT assay during 5 days and compared to the respective isogenic controls. Significance was compared using two-way ANOVA test, (in **f** RPMI8226 $p = 0.0499$, MM.1S $p = 0.0296$; in **g** RPMI8226 sh1 $p = 0.0105$, sh2 $p = 0.0049$;

MM.1 S sh1 $p = 0.0058$, sh2 $p = 0.0021$). Data are represented as mean ± SD from 3 biological replicates. **h–k** $10^7$ of RPMI8226 cells were injected with matrigel subcutaneously into SCID mice ($n = 4$ for each condition). To induce OTUD1 expression (h and i) dox was administered at a dose of 2.5 mg/kg per day intraperitoneally for 3 days every week. All tumor cell lines were stably expressing NanoLuc luciferase for visualization. D-luciferin was injected intraperitoneally 5 min prior to taking bioluminescent images once per week during 5 weeks of the experiment. Photon count (P/sec in i and k) was measured to estimate a tumor size. Significance was compared using two-way ANOVA test (in **i** $p = 0.0346$, in **k** $p = 0.0007$). Data are represented as mean ± SD. **l, m** Tumor xenografts ($n = 4$ biologically independent samples) were mechanically dissociated into single-cell suspension and iIgL content was analyzed by intracellular staining and flow cytometry. Significance was compared using the two-tailed Student's t test (in **l** $p = 0.0003$, in **m** $p = 0.0076$). The whiskers represent minimal and maximal values, the box extends from the 25th to 75th percentiles.

ubiquitinated proteins in myeloma cells belongs to misfolded Igs that are responsible for clogging proteasome.

It is assumed that the unique sensitivity of MM to PIs is caused by a constant proteasome overload[14,15,17,26]. Since our data indicate that OTUD1 activity correlates with the amount of iIgL, ubiquitinated products, and MM patient survival, we hypothesized that changes in the iIgL levels and IgL ubiquitination caused by altered OTUD1 expression might result in a different myeloma response to PIs. As expected, OTUD1 oe (high iIgL) increased the sensitivity of MM cells to all clinically used PIs, while the cells with OTUD1 knock-down (low iIgL) developed profound PI resistance (Fig. 3i, j). Analysis of proteasome expression and activity did not reveal any differences in myeloma cells with OTUD1 oe (Supplementary Fig. 3b–d). Together, these data suggest that OTUD1 modulates MM sensitivity to PIs by regulating a load of misfolded/ubiquitinated Ig on proteasome.

## The E3-ligase KEAP1 binds and ubiquitinates OTUD1

To unveil the mechanism of how OTUD1 potentiates Ig production, we performed a proximity labeling assay in cells with dox-inducible expression of OTUD1 fused to the newest generation of the biotin ligase BirA (TurboID)[27] (Supplementary Fig. 4a). The top potential OTUD1 interactors included two proteins involved in oxidative stress handling: the E3-ligase KEAP1 and the ER-resident peroxiredoxin 4 (PRDX4) (Fig. 4a; Supplementary Data 2).

We explored both KEAP1 and PRDX4 because all Ig molecules are rich in disulfide bonds and their formation is likely accompanied by elevated oxidative stress. First, we validated the OTUD1-KEAP1 interaction by reciprocal co-immunoprecipitation using HA-tagged OTUD1 and Flag-tagged KEAP1 (Fig. 4b). We were able to immunoprecipitate both proteins from MM cells also on the endogenous level (Fig. 4c). A close inspection of the OTUD1 amino acid sequence identified a canonical KEAP1-interaction motif (ETGE)[28] near its N-terminus (Fig. 4d). Deletion of this motif completely disrupted the association of OTUD1 with KEAP1 (Fig. 4e), providing additional evidence for the direct protein-protein binding.

The presence of the KEAP1-substrate ETGE motif indicated that OTUD1 might be ubiquitinated by KEAP1. To test this possibility, we used the OTUD1 C320R construct to avoid self-deubiquitination or aberrant enrichment of ubiquitinated species during extraction[29]. As expected, OTUD1 lacking the ETGE motif was less ubiquitinated compared to the wild type form (Fig. 4f) suggesting that OTUD1 might be a target of the KEAP1 ubiquitin ligase activity. Interestingly, removal of the ETGE sequence did not lead to any changes in OTUD1 protein levels (Fig. 4g). At the same time, none of the tested phenotypes altered upon OTUD1 expression in myeloma cells (Ig production, proliferation, PI sensitivity) was affected by the absence of the KEAP1-binding site in OTUD1 (Supplementary Fig. 4b–d), prompting us to further investigate the role of PRDX4.

## OTUD1 regulates ERAD-mediated PRDX4 degradation

PRDX4 activity is critical for a continuous electron flow in the ER redox cycles, particularly during disulfide bond formation[30]. The appearance of the S-S bridges is the initial and rate-limiting step in the synthesis and folding of Ig molecules[31]. In accordance, elevated levels of PRDX4 were previously associated with the accumulation of Ig in both MM cell lines and primary patient samples[32]. The importance of PRDX4 for Ig production is further highlighted by its unique expression profile that steadily rises during B-cell development and peaks similarly to OTUD1 in bone marrow plasma cells (Supplementary Fig. 5a).

Initially, we validated the OTUD1-PRDX4 interaction in series of reciprocal co-immunoprecipitations with both tagged and endogenous proteins (Fig. 5a, b). Analysis of other members of the PRDX family confirmed the unique selectivity of OTUD1 for PRDX4 (Supplementary Fig. 5b). While OTUD1 is a cytosolic deubiquitinase[33], PRDX4 contains the N-terminal (amino acids 1-38) ER-localization sequence that restricts it to ER lumen[30]. Thus, the direct protein binding is seemingly incompatible. To map and localize the OTUD1-PRDX4 interaction, we used the full length (FL) and the Δ1-38 PRDX4 constructs. Interestingly, OTUD1 failed to immunoprecipitate PRDX4 deletion mutant (Fig. 5c). Additionally, immunofluorescence analysis confirmed colocalisation of FL PRDX4 and OTUD1 at the site of the ER membrane (Fig. 5d, e). Moreover, extraction of subcellular organelles identified OTUD1 and FL PRDX4 but not Δ1-38 PRDX4 present in the ER fractions (Supplementary Fig. 5c, d). Together, these results suggest that localisation of PRDX4 to ER is a prerequisite for its interaction with OTUD1.

Next, we tested whether OTUD1 acts upstream of PRDX4. In vitro, recombinant OTUD1 deubiquitinated PRDX4 indicating that PRDX4 might be an OTUD1 substrate (Fig. 5f). Immunoprecipitation of both Flag-tagged PRDX4 from OTUD1 oe cells (Fig. 5g) and endogenous PRDX4 from MM cells with sh OTUD1 (Fig. 5h) further supported this idea. Additionally, OTUD1 oe increased while OTUD1 knock-down decreased the amount of endogenous PRDX4 protein (Fig. 5i, j) without any effect on the levels of *PRDX4* mRNA (Supplementary Fig. 5e). Finally, cycloheximide pulse-chase confirmed a positive effect of OTUD1 expression on PRDX4 protein levels (Fig. 5k). Thus, we conclude that OTUD1 deubiquitinates PRDX4 and regulates its abundance, possibly by protecting it from degradation.

To date, no activity of ubiquitin ligases was detected inside ER[10]. Therefore, we hypothesized that ubiquitination of PRDX4 might occur during its retrotranslocation to cytosol in the process of ER-associated degradation (ERAD)[34]. In accordance, we detected ubiquitination only of ER-localized FL PRDX4, but not cytosolic Δ1-38 PRDX4 (Fig. 5l). Because proteasome activity is crucial for successful completion of ERAD[35], we tested its effect on PRDX4. As expected, inhibition of proteasome rescued levels of FL PRDX4 to the same extent as

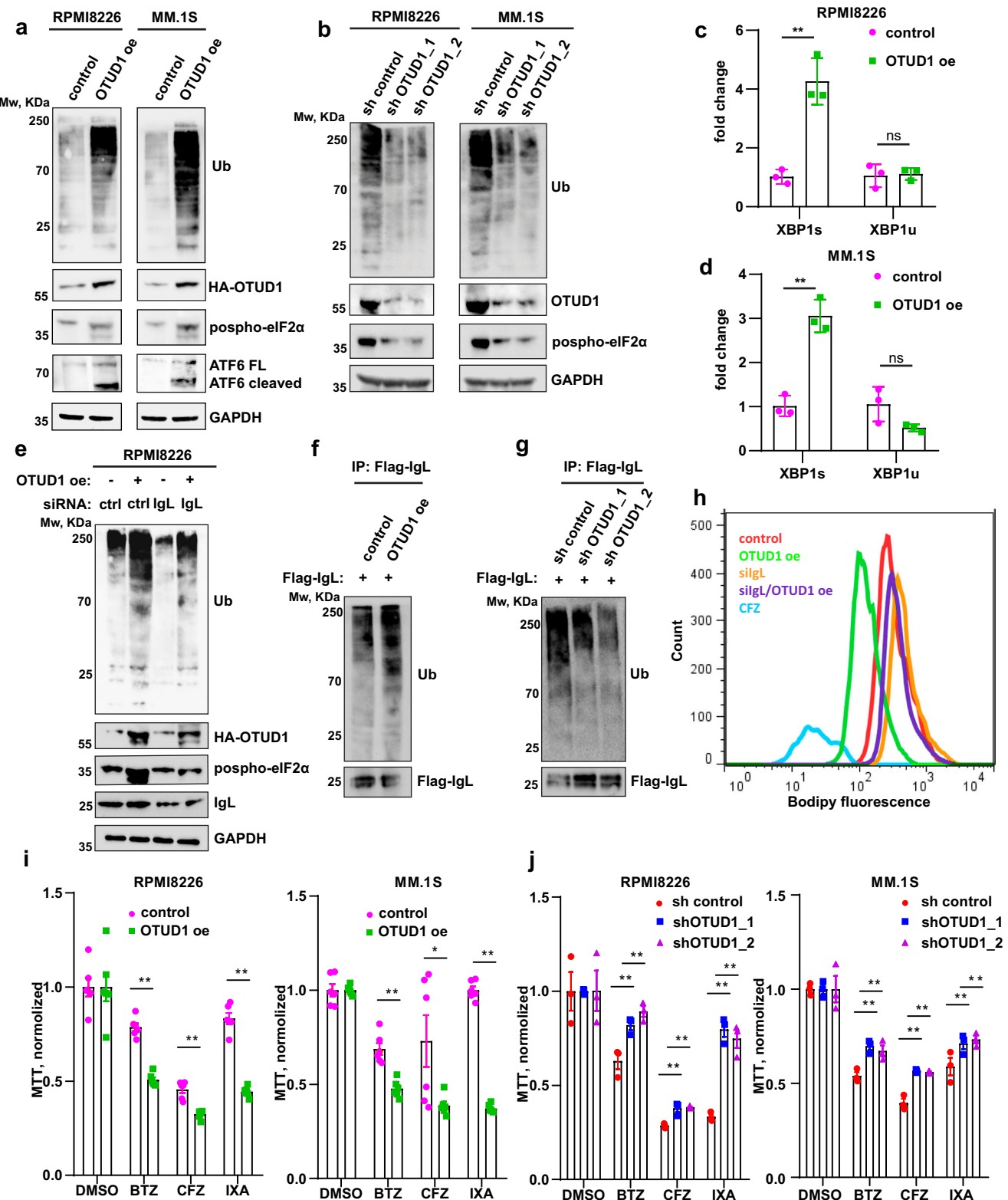

OTUD1 oe (Fig. 5m). On the other hand, blocking the proteasome activity did not affect levels of Δ1-38 PRDX4 (Fig. 5n). Moreover, OTUD1 oe led to a higher amount of FL PRDX4 but not Δ1-38 PRDX4 in the ER fractions (Supplementary Fig. 5c, d), further suggesting that OTUD1 protects PRDX4 from ERAD.

A previous study[36] described OTUD2 (YOD1), a close homolog of OTUD1, as a non-selective regulator of protein retrotranslocation from ER. We analyzed levels of protein disulfide isomerase A6

(PDIA6) in cells with OTUD1 oe in order to examine whether OTUD1 acts in a similar manner. PDIA6 is localized in the ER lumen and, together with PRDX4, participates in the formation of disulfide bonds[37]. Proteasome inhibition promoted PDIA6 levels indicating that PDIA6 is degraded by ERAD. Conversely, OTUD1 oe did not increase PDIA6 protein levels (Fig. 5m). These results advocate OTUD1 selectively regulates the amount of PRDX4 rather than affecting general ERAD mechanisms.

**Fig. 3 | OTUD1 enhances myeloma sensitivity to PIs. a, b** The total amount of ubiquitinated proteins in whole cell lysates from OTUD1 oe (**a**), and sh OTUD1 (**b**) MM cells was analyzed by western blotting. **c, d** The amount of spliced and unspliced form of *XBP1* mRNA was measured by RT-PCR. Significance was compared using the two-tailed Student's t test. (in **c** *XBP1s* p = 0.0025, *XBP1u* p = 0.8216; in **d** *XBP1s* p = 0.0013, *XBP1u* p = 0.0821). Data are represented as mean ± SD from 3 biological replicates. **e** The total amount of ubiquitinated proteins in extracts from RPMI8226 OTUD1 oe cells with and without transfection of si control or si IgL. **f, g** Immunoblot analysis of immunoprecipitated Flag-tagged IgL stably expressed in RPMI8226 cells with OTUD1 oe (**f**) and sh OTUD1 (**g**). **h** Proteasome capacity was analyzed in RPMI8226 cells with OTUD1 oe and siRNA-targeting IgL using intracellular Bodipy fluorescent probe staining and quantified by flow cytometry.

RPMI8226 cells treated with carfilzomib (CFZ) and isogenic non-induced cells were used as reference controls. $5 \times 10^4$ of MM cells with OTUD1 oe (**i**) or sh OTUD1 (**j**) were treated with vehicle (DMSO), 10 nM bortezomib (BTZ), 5 nM carfilzomib (CFZ), and 10 nM ixazomib (IXA) for 16 hrs. Cell viability was estimated by MTT assay and normalized to the control samples. Significance was compared using the two-tailed Student's *t* test. (in **i**: RPMI8226, BTZ p < 0.0001, CFZ p < 0.0001, IXA p < 0.0001; MM1.S, BTZ p = 0.0002, CFZ p = 0.0340, IXA p < 0.0001; in **j**: RPMI8226, BTZ, sh1 p < 0.0001, sh2 p < 0.0001, CFZ, sh1 p < 0.0001, sh2 p < 0.0001, IXA sh1 p < 0.0001, sh2 p < 0.0001; MM1.S, BTZ, sh1 p < 0.0001, sh2 p < 0.0001, CFZ sh1 p < 0.0001, sh2 p < 0.0001, IXA sh1 p = 0.0005, sh2 p < 0.0001). Data are represented as mean ± SD from 6 biological replicates.

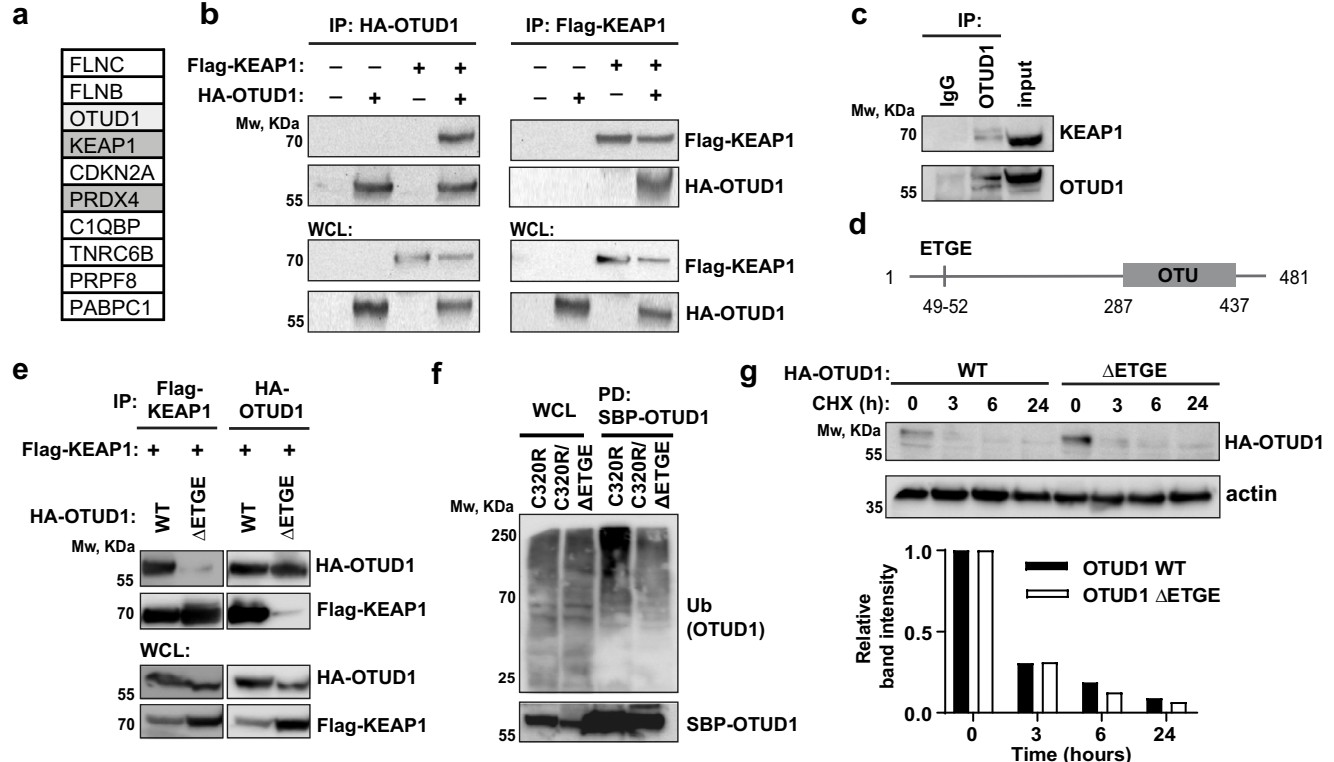

**Fig. 4 | OTUD1 is KEAP1 substrate. a** The list of proteins that were identified as the top 10 OTUD1 interactors by TurboID proximity labeling assay[24] and proteomic analysis. **b** N-terminally HA-tagged OTUD1 and Flag-tagged KEAP1 were immunoprecipitated using anti-HA (left) or anti-Flag (right) agarose resin from whole cell lysates (WCL) of HEK 293 cells and co-immunoprecipitated proteins were analyzed by western blotting. **c** Endogenous OTUD1 was immunoprecipitated from RPMI8226 cells using anti-OTUD1 rabbit polyclonal sera and co-immunoprecipitated endogenous KEAP1 was analyzed by western blotting. **d** The schematic representation of OTUD1 amino acid (AA) sequence highlighting the canonical KEAP1-interaction motif ETGE (AA 49-52) and the catalytic OTU domain (AA 287-437). **e** Flag-tagged KEAP1 and HA-tagged OTUD1 wild type (WT) or ΔETGE mutant were immunoprecipitated using anti-Flag (left) or anti-HA (right) agarose resin from HEK 293 cells and co-immunoprecipitated proteins were analyzed by

western blotting. **f** Ubiquitinated forms of OTUD1 C320R and C320R/ΔETGE N-terminally tagged with Streptavidin-Binding Peptide (SBP) were purified from HEK 293 cells using Streptavidin agarose resin. Total levels of ubiquitin in WCL and ubiquitination of OTUD1 in pull-down (PD) fractions were analyzed by western blotting using anti-ubiquitin antibody. Amount of SBP-OTUD1 both in WCL and PD fractions were detected by Streptavidin conjugated to Horseradish Peroxidase. **g** The degradation rate of HA-tagged OTUD1 WT and ΔETGE mutant was assessed in RPMI8226 cells with doxycycline-inducible expression of OTUD1 upon addition of cycloheximide (50 μg/ml) in the indicated time points. The amount of OTUD1 was determined by western blotting (upper part) and the relative band intensities were quantified using ImageJ software and normalized to β-actin loading control (bottom part).

## OTUD1 mediates Ig production and PI sensitivity through PRDX4

Our data indicate that OTUD1 regulates Ig production, PI sensitivity, and proliferation of MM cells. At the same time, OTUD1 binds and elevates PRDX4 protein amount. Because PRDX4 is an important component of the ER folding machinery, we speculated that changes in the amount of iIgL in myeloma cells with altered OTUD1 expression might be caused by variations in the PRDX4 protein quantity. To test this hypothesis, we rescued the original levels of PRDX4 in OTUD1 oe and sh OTUD1 cells by introducing siRNA targeting PRDX4 (siPRDX4)

or overexpressing PRDX4 (PRDX4 oe), respectively (Fig. 6a, b). As expected, the amount of iIgL restored almost to the basal state upon normalizing the PRDX4 protein amount in both OTUD1 oe and sh OTUD1 cells (Fig. 6c, d). Similarly, increased sensitivity (OTUD1 oe cells) and resistance (sh OTUD1 cells) to bortezomib were reverted in cells with rescued PRDX4 (Fig. 6e, f). Lastly, restoration of PRDX4 levels in cells with both OTUD1 oe and knock-down led to a complete rescue of the total pool of ubiquitinated proteins (Fig. 6g, h). On the other hand, differences in myeloma cell proliferation caused by differential *OTUD1* expression were not changed by normalizing PRDX4 protein

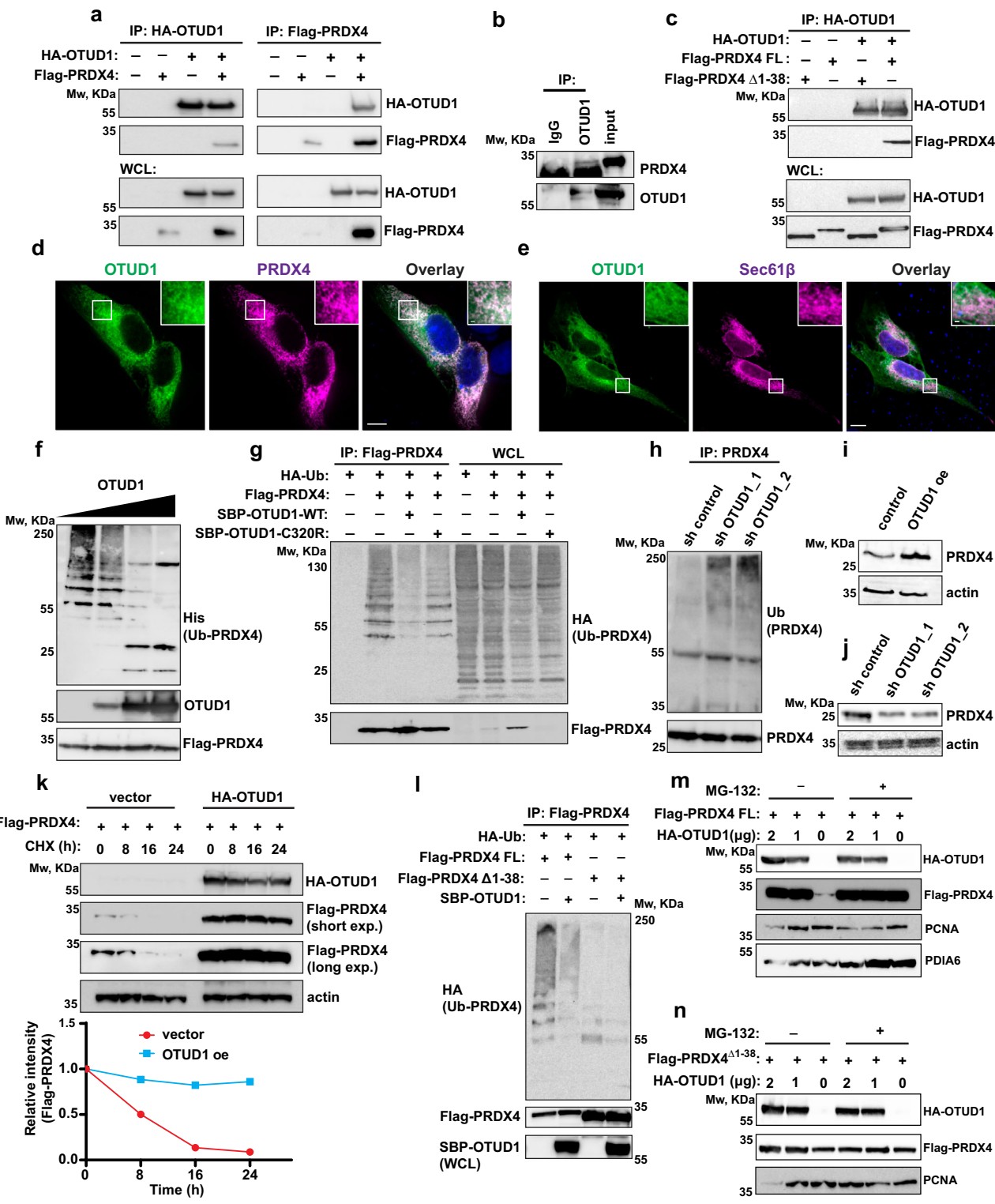

levels or by restoring the IgL levels (Supplementary Fig. 6a–c). Therefore, the effect of OTUD1 on MM cell growth seems to be not dependent on PRDX4 and IgL.

PRDX4 is directly involved in the series of redox reactions required for disulfide bonds formation in the newly formed Ig molecules[30,32]. Therefore, we hypothesized that increase in PRDX4 protein levels might alter the speed of IgL folding creating a disbalance and saturation of the ER folding machinery. Ultimately, amount of misfolded, highly ubiquitinated IgL molecules would rise leading to a proteasome overload. Indeed, PRDX4 oe significantly elevated the amount of ubiquitinated species which was diminished by the introduction of siRNA-targeting IgL (Fig. 6i). Additionally, knock-down of PRDX4 suppressed ubiquitination of IgL (Fig. 6j). Because only misfolded Ig is extracted from ER and ubiquitinated, our data indicate PRDX4 is involved in both Ig synthesis and folding. However, it remained unclear how OTUD1 and PRDX4 activity translates to the altered Ig synthetic rate.

Recent study associated OTUD1 with a ribosome stalling[38]. Moreover, a vast majority of OTUD1-associated proteins in our proteomic analysis belonged to ribosome or translation machinery

**Fig. 5 | OTUD1 controls PRDX4 protein levels. a** N-terminally HA-tagged OTUD1 and Flag-tagged PRDX4 were immunoprecipitated using anti-HA (left) or anti-Flag (right) agarose resin from HEK 293 cells. Co-immunoprecipitated proteins were analyzed by western blotting. **b** Endogenous OTUD1 was immunoprecipitated from RPMI8226 cells using anti-OTUD1 rabbit polyclonal sera and co-immunoprecipitated endogenous PRDX4 was analyzed by western blotting. **c** HA-tagged OTUD1 was immunoprecipitated from HEK 293 cells expressing either full length (FL) or ER-localization sequence deficient (Δ1-38) Flag-tagged PRDX4. Co-immunoprecipitated proteins were analyzed by western blotting. **d, e** Immuno-fluorescence analysis of N-terminally GFP-tagged OTUD1 and N-terminally mApple-tagged PRDX4 (**d**) or N-terminally mApple-tagged SEC61β (**e**) in U2OS cells. Scale bar = 10 μm. **f** Flag-tagged PRDX4 was immunoprecipitated from HEK 293 cells expressing ubiquitin N-terminally tagged with 6xHis (His-Ub) and incubated with increasing concentration of recombinant full length OTUD1. The level of Flag-PRDX4 ubiquitination was analyzed by western blotting using anti-His antibody. **g** Flag-tagged PRDX4 was immunoprecipitated from HEK 293 cells expressing

ubiquitin N-terminally tagged with HA (HA-Ub) and OTUD1 WT or C320R N-terminally tagged with Streptavidin-Binding Peptide (SBP). The ubiquitination pattern was analyzed by western blotting using anti-HA antibody. **h** Endogenous PRDX4 was immunoprecipitated from RPMI8226 cells stably expressing control shRNA or two different shRNA targeting OTUD1. The ubiquitination pattern was detected by western blotting using anti-ubiquitin (Ub) antibody. Whole cell lysates of RPMI8226 cells with OTUD1 oe (**i**) or sh OTUD1 (**j**) were analyzed by western blotting for presence of endogenous PRDX4. **k** Analysis of PRDX4 degradation rate in presence or absence of HA-OTUD1 in cycloheximide pulse-chase assay. The amount of PRDX4 was determined by western blotting (upper part) and the relative band intensities were quantified using ImageJ software and normalized to β-actin loading control (bottom part). **l** Immunoprecipitation of FL PRDX4 and Δ1-38 PRDX4 in presence or absence of SBP-OTUD1. The ubiquitination pattern was analyzed by western blotting using anti-HA antibody. Analysis of protein content of FL PRDX4 (**m**) and Δ1-38 PRDX4 (**n**) in presence of OTUD1 oe and/or proteasome inhibitor MG132.

(Supplementary Data 2) suggesting OTUD1 could mediate translation of *Ig* mRNA. Puromycin incorporation assay revealed that OTUD1 does not affect Ig translation initiation (Supplementary Fig. 6d, e). On the other hand, the overall rate of translation elongation was enhanced in cells with elevated levels of OTUD1 or PRDX4 (Fig. 6k, l). Together these results support the model where OTUD1 increases PRDX4 protein amount which promote the formation of new IgL molecules by enforcing translation elongation. Pronounced Ig synthesis further leads to oversaturation of the ER protein assembly capacity resulting in a raise of both folded and misfolded/ubiquitinated Ig molecules.

**Combining PIs and HSP inhibitors overcomes PI resistance in MM with low iIg**

Finally, to validate the suggested molecular mechanism, we analyzed total ubiquitin levels in aberrant plasma cells isolated from newly diagnosed MM patients who later received PI-based therapy. As in our cell line models, the amount of ubiquitinated proteins directly correlated with the iIgL and iIgH protein content (Fig. 7a top and middle), *OTUD1* expression and PFS (Fig. 7a bottom). Moreover, when we treated several MM cell lines with bortezomib, iIgL levels were significantly lower in the surviving cells compared to the vehicle-treated controls (Fig. 7b). This shows that even within highly homogenous populations of cell lines variations in iIgL correlate with the sensitivity to PIs.

In order to revert the poor prognosis of MM patients with low iIg, we hypothesized that increasing the amount of misfolded, highly ubiquitinated proteins might compensate for the lack of proteasome saturation and re-sensitize the iIg low myeloma cells to PIs. Because accumulation of incorrectly folded proteins is a typical hallmark of ER stress[13], we evaluated a set of the ER stress-inducing drugs that are approved for use in MM patients or are being tested in clinical studies. We applied this drug panel together with bortezomib to PI-resistant sh OTUD1 (iIgL low) myeloma cells (Supplementary Fig. 7a). The best-performing compound for bortezomib co-treatment was the Heat Shock Protein 90 (HSP90) inhibitor tanespimycin (17-N-allylamino-17-demethoxygeldanamycin, 17-AAG) that completely reverted PI resistance in sh OTUD1 cells (Fig. 7c). Analysis of ubiquitinated proteins confirmed the molecular mechanism of tanespimycin-mediated re-sensitization of sh OTUD1 cells to bortezomib was mediated by elevating the amount of ubiquitinated proteins that re-saturated proteasome (Fig. 7d).

## Discussion

Serum M-protein is one of the main diagnostic values measured for every MM patient. Recent reports indicated that the levels and early dynamics of M-protein in response to treatment could serve as an indicator of tumor size and potentially a predictor of PFS[8,39]. However, these studies included a limited number of patients to draw a definitive

conclusion on M-protein prognostic significance. Here, we retrospectively analyzed data from more than 4000 MM patients and provide an evidence that M-protein levels fail to predict patient outcome.

In contrast to M-protein, iIg concentration is likely not affected by tumor burden and thus represents the myeloma Ig production capacity more precisely. Therefore, quantification of iIg from aberrant plasma cells could serve as a better estimate of the Ig synthesis rate. Indeed, we found the concentration of iIg as a robust, independent factor that can be used to anticipate the course of the disease upon induction of PI-based therapy. Previous studies suggested that the changes in the *Ig* expression might have a predictive potential[4,18,26]. However, our results and publicly available data revealed no correlation between the expression of *Ig* mRNA and serum M-protein or iIg. These findings indicate that the control exerted by additional factors on Ig relevant to MM proteostasis occurs mainly at the post-transcriptional level.

In a search for regulators of Ig production we identified deubiquitinase OTUD1 highly expressed in mature plasma cells. Decreased expression of *OTUD1* correlated with poor survival in several unrelated MM patient cohorts. In myeloma cells, OTUD1 negatively affected progression through the cell cycle independently of the Ig expression. This observation is consistent with the previous findings describing OTUD1 as a tumor suppressor in other cancers[40–42]. Additionally, our data support the positive role of OTUD1 in the regulation of Ig production and MM sensitivity to PIs. These phenotypes were fully dependent on ER-localized PRDX4. We found OTUD1 selectively deubiquitinates PRDX4 during retranslocation from ER, thus protecting it from ERAD.

PRDX4 is a critical component of the ER folding machinery where it mediates the formation of intra- and intermolecular disulfide bonds during oxidative protein folding[43]. This process is additionally catalyzed by protein disulfide isomerases (PDIs) and endoplasmic reticulum oxidoreductase 1 (ERO1) which pass the electrons to molecular oxygen generating $H_2O_2$. PRDX4 is responsible for the reduction of peroxide and, to some extent, also PDI enzymes[30,32]. Therefore, PRDX4 determines the assembly speed of disulfide bond-rich Ig molecules. Accordingly, a previous study[44] suggested that the expression of PRDX4 steadily rises during B-cell development and correlates with plasma cells' capacity to produce Ig. However, a precise mechanism how OTUD1-PRDX4 axis mediates the Ig synthesis remained unclear.

Currently emerging evidence indicates on an exciting phenomenon connecting protein folding and translation processivity[45–47]. As for other multidomain proteins, folding of Ig molecules might require ribosome pausing and stalling[48]. Recent study indicated that OTUD1 is able to revert stalled ribosomes[38]. In agreement, our proteomic analysis revealed ribosomal proteins as the most enriched OTUD1 associating proteins. Biochemical assays shown that

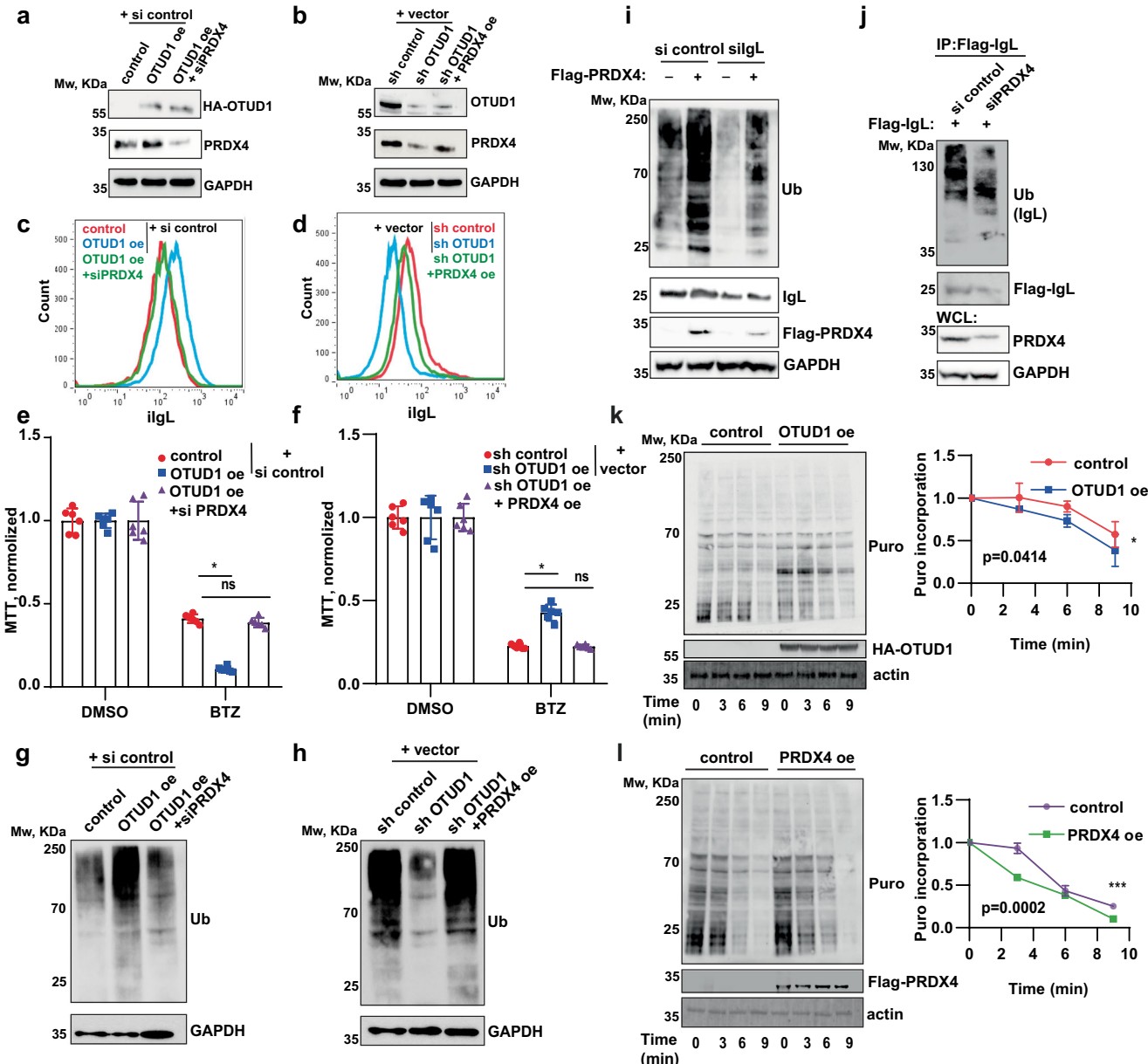

**Fig. 6 | Ig synthesis and PIs sensitivity are controlled by OTUD1-PRDX4 axis.**
**a**, **b** Western blot analysis of endogenous PRDX4 in whole cell lysates from
RPMI8226 cells expressing HA-tagged OTUD1 with or without transfection of
siRNA-targeting PRDX4 (siPRDX4) (**a**) and cells expressing shRNA targeting OTUD1
(sh OTUD1) with or without transfection of PRDX4 (PRDX4 oe) (**b**). **c**, **d** iIgL content
was analyzed by flow cytometry in RPMI8226 cells with OTUD1 oe transfected with
control siRNA or siPRDX4 (**c**) and cells expressing sh OTUD1 with overexpression of
vehicle or PRDX4 oe (**d**). **e**, **f** $5 \times 10^4$ of MM cells with OTUD1 oe with and without
siPRDX4 (**e**) or sh OTUD1 with and without PRDX4 oe (**f**) were treated with 10 nM
bortezomib (BTZ) for 16 h. Cell viability was estimated by MTT assay and normal-
ized to the control samples. Significance was compared using the two-tailed Stu-
dent's t-test (in **e** OTUD1 oe $p < 0.0001$, OTUD1 oe + siPRDX4 $p = 0.1666$; in **f** sh
OTUD1 $p < 0.0001$, sh OTUD1 + PRDX4 oe $p = 0.7597$), ns non-significant. Data are
represented as mean ± SD from 6 biological replicates. **g**, **h** The total amount of
ubiquitinated proteins was analyzed by western blotting in whole cell lysates from

RPMI8226 cells with OTUD1 oe transfected with si control and or siPRDX4 (**g**) and
from cells expressing sh OTUD1 with and without PRDX4 oe (**h**). **i** The total amount
of ubiquitinated proteins in whole cell extracts from RPMI8226 cells with PRDX4 oe
with and without transfection of control siRNA or siIgL. **j** Immunoblot analysis of
immunoprecipitated Flag-tagged IgL stably expressed in RPMI8226 cells trans-
fected with control siRNA or siPRDX4. Ubiquitination pattern was detected with
anti-ubiquitin antibody. **k**, **l** Translation processivity was analyzed in cells trans-
fected with empty vector (control), HA-OTUD1 (**k**) or Flag-PRDX4 (**l**) by blocking
translation initiation with harringtonine and pulsing with puromycin at different
time points. The amount of puromycilated proteins was analyzed by western
blotting and the relative band intensities were quantified using ImageJ software and
normalized to β-actin loading control (right). Significance was compared using two-
way ANOVA test, (in **k** $p = 0.0414$, in **l** $p = 0.0002$). Data are represented as
mean ± SEM from 3 biological replicates.

upregulation of OTUD1 and PRDX4 stimulates translation elongation
and ribosome processivity. This mechanism might potentiate Ig
synthesis without any apparent changes in Ig transcription. Elucida-
tion of how OTUD1 and PRDX4 are linked to protein synthesis will
require further research.

Interestingly, OTUD1 binds and selectively promotes PRDX4
levels leaving other components of the ER folding machinery intact.
Together with increased Ig production, it might create a disbalance
and saturation of the ER redox system resulting in a rise in misfolded/
ubiquitinated forms of Ig. Appearance of defected Ig ultimately

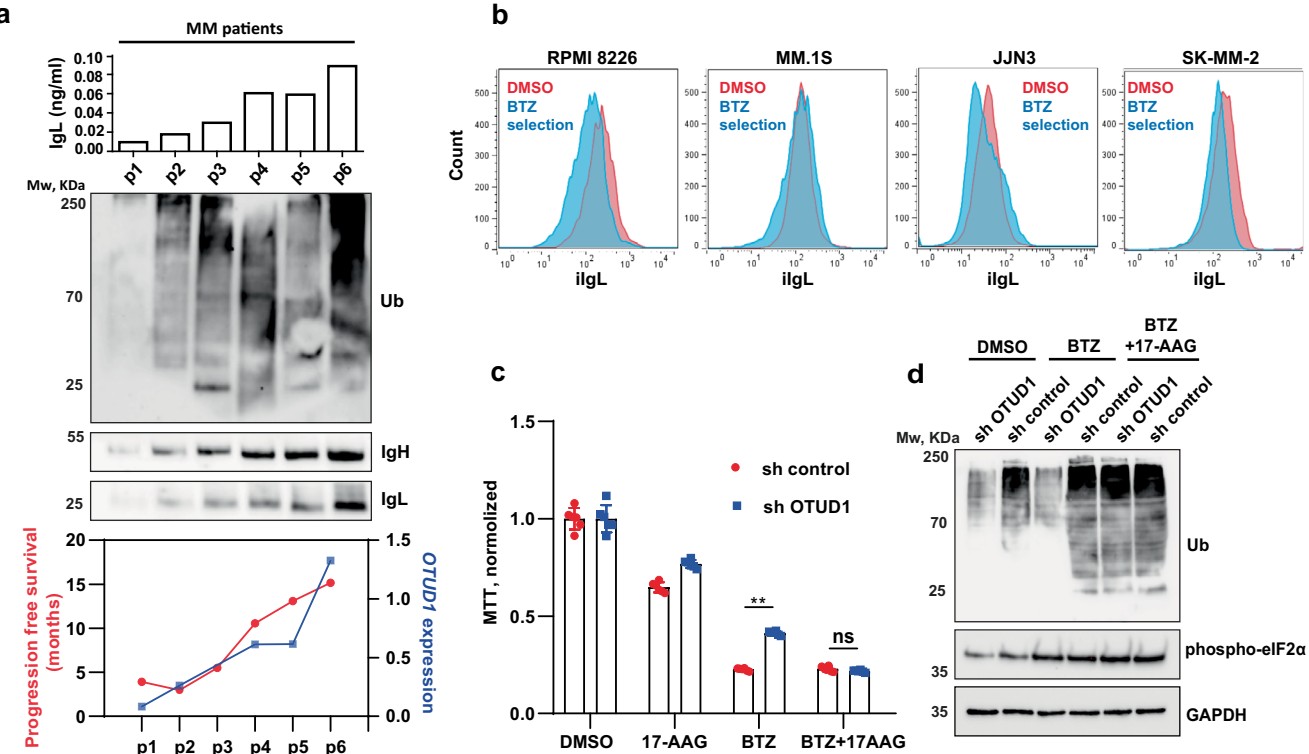

**Fig. 7 | Inhibition of HSP90 overcomes PIs resistance in *OTUD1* low myeloma.** **a** IgL concentration in aberrant plasma cells isolated using specific myeloma panel from newly diagnosed MM patients (*n* = 6) was quantified by ELISA (top graph). Levels of total ubiquitin, IgL, and IgH were detected by immunoblotting in primary cell extracts (western blot). Expression of *OTUD1* in primary cells was analyzed by RT-PCR and plotted together with progression-free survival (PFS) (bottom graph). All 6 patients (p1-p6) received bortezomib (BTZ)-based treatment in the first line of therapy. **b** 10[6] MM cells were treated with 10 nM BTZ or DMSO as control. After 16 h cells were washed with PBS and transferred into fresh media without BTZ. Seventy-two hours later cells were stained with Annexin-V and viable cells were stained with LIVE/DEAD Fixable Blue Dead Cell Stain Kit (Invitrogen). Cells were fixed and permeabilized using FIX & PERM Cell Permeabilization Kit (Invitrogen) and labeled with Lambda-APC-C750™ (Cytognos) antibody. 10[4] events were collected in the LIVE/DEAD negative gate. **c** RPMI8226 cells stably expressing non-mammalian control shRNA or shRNA targeting OTUD1 were exposed to vehicle (DMSO), 10 nM BTZ, 250 nM 17-AAG alone or 10 nM BTZ together with 250 nM 17-AAG for 16 h, and viability was assessed using MTT assay. Significance was compared using the two-tailed Student's *t*-test (BTZ *p* < 0.0001, BTZ + 17-AAG *p* = 0.0669), ns non-significant. Data are represented the mean ± SD from 6 biological replicates. **d** The amount of total ubiquitinated proteins in whole cell lysates from RPMI8226 cells used in (**c**) was determined by western blotting (upper part) and the relative band intensities were quantified using ImageJ software and normalized to GAPDH loading control (bottom part).

augments the ER stress and congests available proteasomes which is reflected by potentiated sensitivity to PIs. In support, a recent study identified *PRDX4* as one of the most highly expressed genes in the primary, PI-sensitive myeloma clones[49]. The relevance of the Ig folding pathway to PI resistance in MM is further emphasized by the development of PDI inhibitors that are currently under investigation as promising agents to boost the efficacy of PIs[50].

In addition to changes in the Ig levels, disturbance of the ER redox circuit might promote oxidative stress that further imbalances proteasome capacity[51]. Interestingly, we identified the E3-ligase KEAP1, a protein sensor of oxidative stress, as an OTUD1 binding partner. Though KEAP1 directly interacts and ubiquitinates OTUD1, this modification did not alter OTUD1 protein content. Additionally, deletion of the canonical KEAP1-binding motif from OTUD1 did not affect all tested, OTUD1-mediated phenotypes in MM cells. Thus, the biological relevance of the KEAP1-OTUD1 complex remains to be elucidated.

In summary, our work unveils the OTUD1-PRDX4 axis as a yet undescribed regulatory pathway driving Ig production and PI sensitivity in MM cells and highlights importance of OTUD1 in myeloma proliferation (Fig. 8). We suggest that expression of *OTUD1* and, particularly, iIg concentration could be considered as promising prognostic and stratification parameters for newly diagnosed MM patients. Since the amount of iIg can be measured directly, future studies should

identify precise and clinically relevant thresholds to recognize PI-responsive and non-responsive patients that would profit the most from PI-based combination therapy, especially in the era of PI-free standard of care regimens.

## Methods

### MM cell lines

Human RPMI8226, MM.1 S, HEK 293, and U2OS cell lines were obtained from American Type Culture Collection under respective cat. n. CCL-155, CRL-2974, CRL-1573, HTB-96. JJN-3, SK-MM-2 were obtained from Deutsche Sammlung von Mikroorganismen und Zellkulturen under respective cat. n. ACC 541, ACC 430. 293FT cells were obtained from Invitrogen under cat. n. R70007. Cell lines were maintained in RPMI1640 (for RPMI8226, MM.1S, JJN-3 and SK-MM-2) or DMEM (for HEK 293, U2OS, 293FT) medium containing 10% heat-inactivated fetal bovine serum.

### Data collection for PFS and OS analysis

Analysis of clinical data was retrospective using data from the Czech Registry of Monoclonal Gammopathies (RMG: https://rmg.healthregistry.org/). All patients signed informed consent for data collection and the study protocol prior entering the RMG. The consent form and the study protocol have been approved by the ethical committee of the University Hospital of Ostrava. Parameters of interest

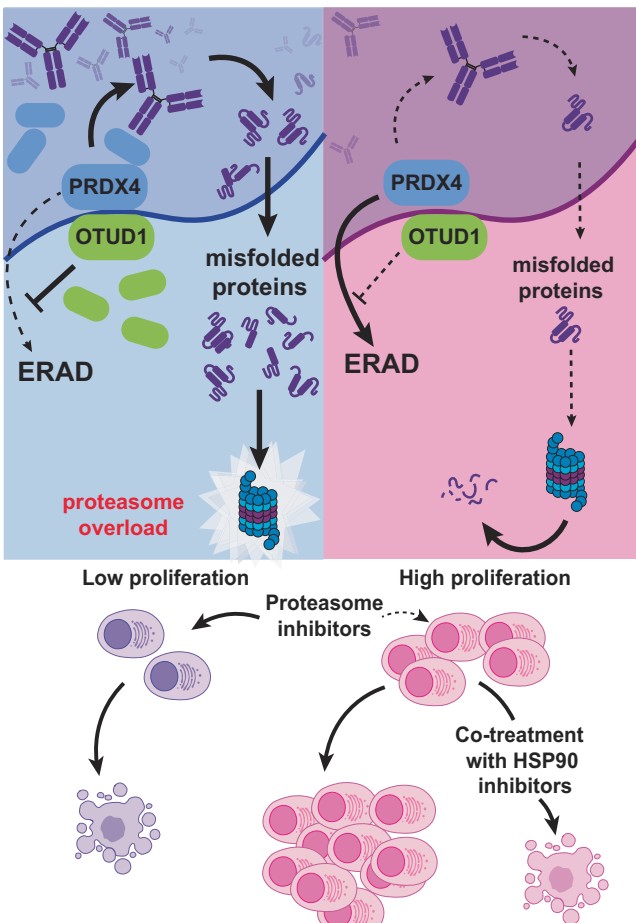

**Fig. 8 | Model of OTUD1-PRDX4 axis regulating Ig production in myeloma cells.** Suggested mechanistic model of OTUD1 driving MM cells proliferation, Ig folding and PIs sensitivity via modulation of PRDX4 levels together with the proposed rational combination treatment to re-sensitize OTUD1 low / iIg low MM cells to PIs. Created with Biorender.com.

in the Registry contain all demographic data, disease characteristics, and treatment intervals including overall survival (OS) and progression-free survival (PFS).

## Statistical analysis of PFS/OS cohorts
Data were described by absolute and relative frequencies of categorical variables and median (min−max) of continuous variables. Survival analysis (PFS and OS) was computed by Kaplan-Meier method and statistical significance of differences in survival among subgroups was assessed using the log-rank test. All statistical tests were performed at a significance level of $p = 0.05$ (all tests two-sided). Analysis was performed in the SPSS software (IBM Corp. Released 2017. IBM SPSS Statistics for Windows, Version 25.0.0.1 Armonk, NY: IBM Corp.) and software R version 4.0.1. (www.r-project.org). Cubic spline analysis was performed using rms v6.2-0R package and R v4.0.5.

## Human primary MM samples
All patients gave a written informed consent before sample collection. The collection and the study protocol were approved by the Ethical Committee of the University Hospital Ostrava. Fresh bone marrow aspirates from MM patients ($n = 106$) were routinely collected in the Haematooncology department of the University Hospital Ostrava between 2013 and 2019. Malignant plasma cells were isolated using CD138 MicroBeads (Miltenyi Biotec) by autoMACS Pro Separator.

## Mice
In total 16 SCID mice (CB17/Icr-Prkdc-scid/IcrIcoCrl, female, 7-12 weeks, The Jackson Laboratory) were used for in vivo experiments (4 mice per condition). All animal experiments were approved by the Animal Ethics Committee of the Faculty of Medicine, Ostrava University and the Animal Ethic Board of the Ministry of Education, Youth and Sport of the Czech Republic n. MSMT-4072/2021-3.

## In Vivo Experiments
$10^7$ RPMI8226 cells expressing NanoLuc in 0.2 ml PBS (1:1 matrigel) were injected subcutaneously into an anesthetized mouse. Tumor size was analyzed by bioluminescence starting 14 days after inoculation. To measure bioluminescence D-luciferin was injected intraperitoneally 5 min prior to imaging once per week during 5 weeks of the experiment. Photon count (P/s) was measured analyzed using Bruker MI SE 7.2 software to estimate a tumor size. To induce OTUD1 expression, doxycycline was administered at a dose of 2.5 mg/kg per day intraperitoneally for 3 days every week. After sacrifice, tumor xenografts were mechanically dissociated into single-cell suspension and iIgL content and GFP expression were analyzed by flow cytometry. Maximal tumor size approved by the ethical committee was 18 mm in diameter and this size was not exceeded in any of the animal.

## Surface phenotype of aberrant plasma cells
Bone marrow aspirates were diluted in the Red Blood Cell lysis buffer (155 mM $NH_4Cl$, 12 mM $NaHCO_3$, 0.1 mM EDTA; 1 ml: 10 ml ratio) and incubated at room temperature for 15 min. Aberrant plasma cell phenotype was determined according to the EuroFlow protocol (stained for CD38, CD138, CD45, CD56, CD117, CD27). Cells were sorted using BD FACSAria II and processed immediately using. Flow cytometry data were acquired by BD FACSDiva™ Software v9.0 and CytExpert software v2.4; and analyzed by FlowJo v10.

## ELISA
Aberrant plasma cells were lysed in RIPA buffer containing protease inhibitor cocktail (Roche), iodoacetamide (50 µM) and N-ethylmaleimide (10 mM). Protein concentration was measured using BCA Protein Assay Kit (Pierce) and cell lysates were diluted to 2 µg/ml of total protein. Concentration of iIgL was determined by Human Lambda or Human Kappa ELISA kit (Bethyl Laboratories).

## Quantitative real-time polymerase chain reaction
Total RNA was extracted from cells using the RNeasy Mini Kit (Qiagen). The RNA aliquots were stored at −80 °C. The quality (purity and integrity) of the RNA samples was assessed using the Agilent 2100 Bioanalyser with the RNA 600 NanoLabChip reagent set (Agilent Technologies). The RNA was quantified using a spectrophotometer. Complementary DNA (cDNA) synthesis was performed using the RevertAid First Strand cDNA Synthesis Kit (Thermo Scientific) according to the manufacturer's instructions. Quantitative RT-PCR was conducted using PowerUp™SYBR™ Green Master Mix (Applied Biosystems) on StrepOnePlus Real-Time PCR System (Applied Biosystems). Relative mRNA expression was calculated by 2−ΔΔCt method and normalized to *GAPDH* or *GUSB* gene. Oligonucleotide sequences used in the study can be found in Supplementary Data 3.

## Lentiviral transduction
Lentiviral constructs (1.64 pmol, pLKO.1 for shRNA and pCW57.1 for overexpression) were used for plasmid construction and transfected into 293FT cells ($3.5 \times 10^6$ cells seeded in 10 cm dish overnight) together with the helper plasmids (0.72 pmol of pMD2.G and 1.3 pmol of psPAX2) using jetPRIME® transfection reagent (Polyplus transfection). Viral supernatants were collected 48 h post transfection, mixed with PEG-8000 solution (final concentration 10 % W/V) and sodium chloride (final concentration 0.3 M) and agitated overnight at 4 °C. Next day,

viral particles were concentrated by spinning down at 1600 g for 60 min (4 °C). Resulted pellet was resuspended in PBS. $5 \times 10^5$ RPMI8226 or MM.1S cells were incubated with viral supernatant in presence of 10 μg/mL polybrene (Sigma-Aldrich) in a final volume of 2 mL and spin-infected for 1 h at 900 g (34 °C). Cells were then supplemented with 3 mL fresh medium, continued culture for at least 48 h and selected with puromycin (2 μg/ml) overnight.

### Intracellular IgL analysis by flow cytometry
$5 \times 10^5$ RPMI8226 or MM.1S cells were stained with LIVE/DEAD™ Fixable Blue Dead Cell Stain Kit (Invitrogen). Cells were fixed and permeabilized using FIX & PERM™ Cell Permeabilization Kit (Invitrogen) and labeled with Lambda-APC-C750™ (Cytognos) antibody. $10^4$ events were collected in LIVE/DEAD negative gate, MFI was analyzed using geometric mean statistics. Flow cytometry data were acquired by BD FACSDiva™ Software v9.0 and CytExpert software v2.4; and analyzed by FlowJo v10. Gating strategy is shown in the Supplementary information file. The gating strategy is shown in Supplementary Fig. 8.

### Proliferation and viability assay
$10^4$ RPMI8226 or MM.1S cells were seeded per one well in 96-well plate in triplicates for each timepoint. Cell proliferation was measured by MTT assay at days 1–5. For viability assay $5 \times 10^4$ were seeded, treated with a drug or combination of drugs and analyzed in 16 h. In the MTT assay, cells were incubated with the MTT labeling reagent (final concentration 0.5 mg/ml) for 30 min at 37 °C, 5% $CO_2$, formazan crystals were solubilized in DMSO and absorbance was measured at 540 nm.

### Proteasome saturation assay
RPMI8226 cells were collected in proteasome activity assay buffer (50 mM TRIS, pH 7.5, 10 mM NaCl, 250 mM sucrose, 5 mM $MgCl_2$, 1 mM EDTA, 1 mM DTT, and 2 mM ATP) and lysed by sonication. Lysates were centrifuged at 17,000 g for 10 min at 4 °C. Protein concentration was determined using the BCA protein assay (Thermo Fisher Scientific). 25 μg of total protein of cell lysates was transferred to a 96-well black plate (Corning Costar) and then the fluorogenic Suc-LLVY-AMC (100 μM, Enzo) substrate was added to lysates. Fluorescence intensity (340 nm excitation, 440 nm emission) was monitored using an Infinite F Plex (Tecan) every 20 min for 1.5 h at 37 °C and the data were analyzed by GraphPad Prism. Gating strategy is shown in the Supplementary information file.

### TurboID proximity labeling assay
$2.5 \times 10^7$ HEK 293 cells with doxycycline-inducible expressing of TurboID N-terminally fused to OTUD1 and TurboID only were treated with biotin (50 μM) for 1 h before lysis with Urea buffer (8 M Urea, 50 mM TRIS, pH 7.6, 1 mM DTT) supplemented with 1% Triton X-100 and protease inhibitor cocktail (Thermo Fisher Scientific). After 30 min incubation on ice, the concentration of Triton X-100 was decreased to 0.5% by dilution with Urea buffer. The cells were further sonicated and lysates were clarified by centrifugation at 17,000 g for 10 min at 4 °C. Protein concentration was determined by BCA protein assay (Thermo Fisher Scientific). The lysates were mixed with 25 μL of Streptavidin sepharose high performance resin (Cytiva) and incubated on a rotator at 4 °C overnight. Beads were washed extensively with Urea buffer and transferred to new tubes before washing with ammonium bicarbonate buffer (1 mM biotin, 50 mM ammonium bicarbonate). The enriched biotinylated proteins were subjected to on-bead trypsin digestion (1 μg of trypsin at 37 °C overnight). The digested peptides were collected and desalted using in-house made StageTips packed with C18 disks (Empore) before mass spectrometry analysis.

### Mass spectrometry (MS) and proteomic data analysis
Peptides were separated and analyzed on an UltiMate 3000 RSLCnano system coupled to an Orbitrap Fusion Tribrid mass spectrometer (both

from Thermo Fisher Scientific). Peptides were firstly loaded onto an Acclaim PepMap300 trap column (300 μm × 5 mm) packed with C18 (5 μm, 300 Å, Thermo Fisher Scientific) in loading buffer (0.1% trifluoroacetic acid in 2% acetonitrile) for 4 min at 15 μL/min and then separated in an EASY-Spray column (75 μm × 50 cm) packed with C18 (2 μm, 100 Å, Thermo Fisher Scientific) at a flow rate of 300 nL/min. Mobile phase A (0.1% formic acid in water) and mobile phase B (0.1% formic acid in acetonitrile) were used to establish a 60-min gradient from 4% to 35% B. Eluted peptides were ionized by electrospray. A full MS spectrum (350-1400 $m/z$ range) was acquired at a resolution of 120,000 at $m/z$ 200 and a maximum ion accumulation time of 100 ms. Dynamic exclusion was set to 60 s. Higher-energy collisional dissociation (HCD) MS/MS spectra were acquired in iontrap in rapid mode and normalized collision energy was set to 30% with maximum ion accumulation time of 35 ms. The automatic gain control for MS and MS2 was set at 1E6 and 5E4, respectively. Top speed mode with 2 s cycle time and lower intensity threshold 5E3 were selected. An isolation width of 1.6 $m/z$ units was used for MS. All raw data were processed and searched using MaxQuant 1.6.3.4[52] with the UniProtKB reviewed human protein database (release 2020_07; 20,381 sequences). Trypsin specificity was set C-terminally to arginine and lysine residues, also allowing the cleavage of proline bonds. Two missed cleavage sites of trypsin were allowed. Carbamidomethylation of cysteine was selected as fixed modification and N-terminal protein acetylation and methionine oxidation as variable modifications. The false discovery rate of both peptide identification and protein identification was set as 1%. The options of "Second peptides" and "Match between runs" were enabled. Label-free quantification was used to quantify the difference in protein abundance between different samples[53]. Data analysis was performed using Perseus 1.6.1.3 software[54] and significance was compared using two-sided Student t-test (Supplementary Data 2). The obtained dataset was further correlated with the published OTUD1 interaction data[55].

### Recombinant protein production and purification
The full coding sequence of the human *OTUD1* gene (gene ID 220213) was cloned into a pGEX-6P-1 vector containing N-terminal 6His-GST tag cleavable by tobacco etch virus (TEV) protease. The cloned gene was expressed in BL21(DE3) RIPL strain of *E. coli*. Cells were grown in LB medium at 37 °C until $OD_{600}$ of 0.6 after which the expression was induced by 0.5 mM IPTG. Cells were grown for another 18 h at 18 °C, pelleted by centrifugation, and resuspended in GST-binding buffer (50 mM TRIS, pH 7.5, 500 mM NaCl). Cell suspensions were supplemented with lysozyme (1 mg/mL), PMSF (1 mM), $MgCl_2$ (1 mM) and benzonase (10 mUnits/μL), and sonicated. Bacterial lysates were obtained by centrifugation for 30 min at $12,000 \times g$. 6His-GST tagged proteins were captured on a GSTrap 5 ml FF column (Cytiva) and eluted with 20 mM glutathione. 6His-GST-OTUD1 was further subjected to 6His-TEV protease cleavage at 4 °C overnight. 6His-GST tag with 6His-TEV protease was then captured using immobilized metal affinity chromatography on a HisTrap column (Cytiva), whereas purified recombinant OTUD1 protein was present in the flow-through fractions. The purified OTUD1 was concentrated and finally exchanged into assay buffers using 7 kDa molecular mass cut-off Zeba spin desalting columns (Thermo Fisher Scientific). TEV protease 6His-TEV(S219V)−5Arg was prepared in house following the modified method from ref. 56.

### Co-Immunoprecipitation of endogenous proteins
Co-immunoprecipitation of OTUD1 and PRDX4 complexes was performed by lysing RPMI8226 cells in immunoprecipitation buffer (20 mM TRIS, pH 7.5, 150 mM NaCl, 0.5% NP40) containing phosphatase and protease inhibitor cocktail (Thermo Fisher Scientific), 20 mM N-ethylmaleimide, and 5 mM iodacetamide for 1 h on ice. Lysates were clarified by centrifugation at 17,000 g for 10 min at 4 °C and pre-cleared by incubation with Protein A/G PLUS-agarose resin (Santa Cruz Biotechnology) on a rotator for 1 h at 4 °C. Then, 4 mg of lysate was

incubated with 4 µg of anti-PRDX4 rabbit polyclonal antibody (10703-1-AP, Proteintech) or with anti-OTUD1 rabbit polyclonal sera (Moravian Biotechnology) at 4 °C overnight. The antibody-bound protein complexes were incubated with Protein A/G PLUS-agarose beads (Santa Cruz Biotechnology) on a rotator for 4 h at 4 °C. After washing the beads with immunoprecipitation buffer, bound proteins were eluted by boiling in NuPAGE LDS sample buffer (Invitrogen) and analyzed by western blotting.

## Co-Immunoprecipitation of overexpressed proteins

HEK 293 cells expressing N-terminally HA-tagged (OTUD1) or Flag-tagged (KEAP1, PRDX4) proteins were harvested, washed with ice-cold PBS, and lysed in immunoprecipitation buffer (20 mM TRIS, pH 7.5, 150 mM NaCl, 0.5% NP40) containing protease inhibitor cocktail (Thermo Fisher Scientific), 20 mM N-ethylmaleimide, and 5 mM iodacetamide for 30 min on ice. The lysates were clarified by centrifugation at 17,000 × $g$ for 10 min at 4 °C and mixed with anti-FLAG M2 or anti-HA agarose resin (both from Sigma Aldrich). After 2 h incubation on a rotator at 4 °C, beads were washed extensively with immunoprecipitation buffer. Bound proteins were eluted by excess of FLAG or HA peptide (0.4 mg/mL, both from Sigma Aldrich) and analyzed by western blotting or used for in vitro deubiquitination assay.

## Cycloheximide pulse-chase assay

$2 \times 10^6$ of RPMI8226 cells were treated with 50 µg/mL of cycloheximide for indicated time points. Afterwards, cells were washed with cold PBS and lysed in RIPA buffer. Cell lysates were analyzed by immunoblot. Densitometry analysis of immunoblots was performed using ImageJ v1.49 (National Institutes of Health).

## In vitro deubiquitination assay

HEK 293 cells expressing N-terminally Flag-tagged PRDX4 and 6His-ubiquitin were lysed and Flag-PRDX4 was immunoprecipitated as described above. Next, Flag-PRDX4 protein was mixed with increasing concentration of purified recombinant OTUD1 protein (0.1, 1, and 2 µM) in the presence of DTT (0.5, 5, and 10 mM, respectively) in the in vitro deubiquitination assay buffer (50 mM TRIS, pH 7.5, 150 mM NaCl). The deubiquitination reactions were carried out at 37 °C for 30 min, quenched by boiling in NuPAGE LDS sample buffer (Invitrogen), and analyzed by western blotting.

## Pull down of ubiquitinated proteins

HEK 293 cells expressing N-terminally Streptavidin-Binding Peptide (SBP)-tagged OTUD1 C320R and C320R/∆ETGE mutants were lysed as for immunoprecipitation experiments. Next, ubiquitinated proteins were enriched using Ubiquitin pan Selector agarose resin (NanoTag Biotechnologies). After 2 h incubation on a rotator at 4 °C, beads were washed with immunoprecipitation buffer and bound proteins were eluted by boiling the beads in NuPAGE LDS sample buffer (Invitrogen) and analyzed by western blotting.

## SDS-PAGE and western blotting

The immunoprecipitates and cell lysates were resolved by SDS-PAGE and transferred to PVDF membrane. Membranes were blocked in 5% (w/v) non-fat milk (Roth) in PBS-T (phosphate buffer saline, 0.05% Tween-20) and incubated overnight at 4 °C in 1% (w/v) BSA/PBS-T with the appropriate primary antibodies. Primary antibodies used at indicated dilutions include: anti-β-Actin (3700S, clone 8H10D10, CST, 1:500), anti-FLAG (F1804, clone M2, Sigma Aldrich, 1:1000), anti-GAPDH (97166S, D4C6R, CST, 1:1,000), anti-HA (11867423001, clone 3F10 Roche, 1:1000), anti-human IgG (H + L) (SAB3701329, Sigma Aldrich, 1:500), anti-human Ig lambda (CYT-LAC750, Cytognos, 1:1000), anti-KEAP1 (8047S, D6B12, CST, 1:1,000), anti-OTUD1 (custom made rabbit polyclonal sera, Moravian Biotechnology), anti-polyHistidine (H1029, clone HIS1 Sigma Aldrich, 1:1,000), anti-PRDX4

(60286-1-Ig, Proteintech, 1:1000), anti-PRDX4 (19178-1-AP, Proteintech, 1:1000), anti-PSMB5 (19178-1-AP, Proteintech, 1:1000), anti-PSMC6 (A303-825A, Bethyl Laboratories, 1:1,000), anti-PSMA2 (sc-377148, clone B4, SantaCruz, 1:1000), anti-PSMB1 (sc-374405, clone D9, SantaCruz, 1:1000), anti-PSMB7 (sc-365725, clone H3, SantaCruz, 1:1000), anti-PCNA (13110S, clone D3H8P, CST, 1:1000), anti-PDIA6 (A304-519A, lot n. 1, Bethyl, 1:1000), anti-calnexin (sc-46669, clone E10, SantaCruz, 1:1000), anti-ATF6 (65880S, clone D4Z8V, CST, 1:1000), anti-puromycin (MABE343, clone 12D10, Sigma Aldrich), anti-phospho-eIF2α (Ser51) (3398, clone D9G8, CST, 1:1000), anti-ubiquitin (3936S, clone P4D1, CST, 1:1000). Membranes were subsequently washed with PBS-T and incubated with HRP-conjugated secondary antibodies for 1 h at room temperature. HRP-coupled secondary antibodies used at indicated dilutions include: goat anti-rabbit-IgG (111-035-144, Jackson ImmunoResearch, 1:2000), goat anti-mouse-IgG (115-035-146, Jackson ImmunoResearch, 1:2000), goat anti-rat-IgG (NA935, Cytiva, 1:2000), streptavidin (016-030-084, Jackson ImmunoResearch, 1:10,000). After further washing, signal detection was performed using ECL (Thermo Fisher Scientific) and ChemiDoc MP System (Bio-Rad). ImageJ v1.49 (National Institutes of Health) was used to analyze protein bands by densitometry. Custom made rabbit polyclonal anti-OTUD1 antibody was validated for use in western blotting on lysates from human wild type and OTUD1 knockout cell lines and also using recombinant human full length OTUD1 protein.

## Cell fractionation

$3 \times 10^7$ HEK 293 cells were washed in PBS and resuspended in hypotonic buffer (10 mM HEPES, pH 7.8, 1 mM EGTA, 25 mM potassium chloride) in 3X volume of the cell pellet. After 20 min of incubation at 4 °C cells were centrifuged, resulted pellet was resuspended in isotonic buffer (10 mM HEPES, pH 7.8, 250 mM sucrose, 25 mM potassium chloride, 1 mM EGTA) in a 4X volume of the new cell pellet. Cells were lysed by passing through 27-gauge needle 4 times. The lysate was centrifuged at 1000 × $g$ for 10 min at 4 °C. Supernatant was transfered to a new tube and centrifuged ar 12,000 × $g$ for 15 min at 4 °C. Supernatant was transferred to 15 ml tube, 7.5 volumes of cold 8 mM CaCl were slowely added while constantly vortexed. The tube was rotated for 15 minutes at 4 °C and centrifuged at 8000 × $g$ for 10 min at 4 °C. Supernatant containing cytoplasmic fraction was concentrated to 50 µl using Amicon® Ultra-4 10 K Centrifugal Filter Device. The pellet contained precipitated ER, it was washed in isotonic buffer at 8000 × $g$ for 10 min at 4 °C and resuspended in 30 µl.

## Bodipy staining

RPMI8226 cells were incubated in growth media with 500 nm of Me4BodipyFL-Ahx3-L3-VS probe for 1 h at 37 °C. Cells were washed in PBS and analyzed by flow cytometry. As a negative control cells were treated with 1 µM of carfilozomib for 1 h before staining. Flow cytometry data were acquired by BD FACSDiva™ Software v9.0 and CytExpert software v2.4; and analyzed by FlowJo v10. Gating strategy is shown in the Supplementary information file. The gating strategy is shown in Supplementary Fig. 8.

## Fluorescence microscopy

For microscope observations, the cells were maintained overnight in 24-well plate with the glass cover slip in and then transiently co-transfected with OTUD1-GFP and PRDX4-mApple or mApple-Sec61β constructs using PEI. After 48 h from transfection, the cells were washed in PBS, fixed with 4% PFA in PBS for 15 min at room temperature (RT) and washed in PBS. The residual aldehydic groups were reduced by 15 min incubation at RT in 0.1 M glycine. Subsequently, samples were washed in PBS and stained with (4′,6-diamidine-2′-phenylindole dihydrochloride) DAPI for 5 min at RT and washed 2 times in PBS. Washed slides were let to air dry and mounted in a mounting medium. All samples were examined with a Leica Dmi8 platform.

Samples were documented using a 63×/1.40 NA (numerical aperture) oil immersion objective.

## Designing and selection of guide RNA

The CHOPCHOP web tool (https://chopchop.cbu.uib.no/) was used to design sgRNAs. The sequences targeting the first or second exon were chosen. In case of multiple isoforms, sequences present in the common exon or the canonical isoform were selected. Sequences with the GC content higher than 45%, efficiency score higher than 0.5, low self-complementarity, and mismatch score were preferred.

## Generation of knockout cell lines

For generating gene knockout in RPMI8226, CRISPR-Cas9 technology based on two vector system was used. Plasmids for transfection were purchased from Addgene (Lenti-guide puro #52963 and Lenti-cas9 blast #52962). Three sgRNA per gene were simultaneously applied. For lentivirus generation, $1 \times 10^6$ 293FT cells was seeded in 6 well plate and transfected with the respective plasmids the next day. Packaging plasmids (psPAX2, 2.58 μg; pMDG, 780 ng), 3 Lenti-guide puro vectors (1 μg each) containing sgRNA to specific gene and 11 μL of linear polyethylenimine (MW 25 000, 1 mg/ml) were mixed in 450 μl of Opti-MEM (Gibco) for 15 min and added to the cells. Media was changed the next day. After 72 h media was collected and mixed with PEG-8000 solution (final concentration 10 % W/V) and sodium chloride (final concentration 0.3 M) and agitated overnight at 4 °C. Next day, viral particles were concentrated by spinning down at 1,600 g for 60 min (4 °C) and resulted pellet was resuspended in PBS. $1.5 \times 10^5$ RPMI8226 cells stably expressing Cas9 were mixed with lentiviral particles and 10 μg/ml of Polybrene (Millipore), and spin-infected for 1 h at $900 \times g$ (34 °C) in 96-well round bottom plate. The cells were then seeded into 24-well plate, and the media was changed the next day. After 72 h post infection, cells were selected with puromycin (2 μg/ml). Oligonucleotide sequences used in the study can be found in Supplementary Data 3.

## SUnSET assay

RPMI8226 cells were incubated with 10 μg/ml of puromycin for 10 min at 37 °C and washed in PBS and lysed in immunoprecipitation buffer (20 mM TRIS, pH 7.5, 150 mM NaCl, 0.5% NP40). Puromycin incorporation was analyzed by western blot.

## Translation processivity analysis (SunRISE assay)

HEK 293 cells were treated with 2 μg/ml of harringtonine to block translation initiation and after 0, 3, 6 and 9 min were pulsed with 10 μg/ml of puromycin for 10 min at 37 °C, washed in PBS and lysed in immunoprecipitation buffer (20 mM TRIS, pH 7.5, 150 mM NaCl, 0.5% NP40). Puromycin incorporation was analyzed by western blot. Densitometry analysis of immunoblots was performed using ImageJ v1.49 (National Institutes of Health).

## Statistical analysis and Reproducibility

The statistical significance of differences between various groups was calculated with the two-tailed paired t-test or ANOVA, and error bars represent standard deviation of the mean (SD). Statistical analyses, unless otherwise indicated, were performed using GraphPad Prism 5. Data are shown as mean ± SD. Multivariate Cox proportional hazard model was computed using R 4.0.3 and survival v3.2.11 package. Images of gels in the figures are representative experiments which have been repeated in form of independent biological replicates multiple times as indicated. Figures 2b, c, 3a, b, 6a, b, g, h, k, l ($n = 3$); 3e, 4b, c, e, g, 5a–c, f–h, k–n, 6i, j and Supplementary Figs. 3a, c, 5b–d, 6d, e ($n = 2$); 7a ($n = 1$). Source data are provided as a Source Data file.

## Reporting summary

Further information on research design is available in the Nature Portfolio Reporting Summary linked to this article.

## Data availability

The raw data related to the patient cohort are deposited in the Czech Registry of Monoclonal Gammopathies and are available under restricted access; access can be obtained upon request. The raw proteomic data were deposited in the PRIDE database under accession number: PXD037309. Microarray and survival data for MM patients from Goswami et al., 2013 and Myeloma Institute for Research and Therapy, Donna D. and Donald M. Lambert Laboratory of Myeloma Genetics are publicly available from GEO database under accession numbers GSE2658 and GSE4581 respectively. Gene expression data in B-cell lineage from Jourdan et al., 2014 is available at http://www.genomicscape.com/microarray/data_management.php?view=2. Gene expression in human immune cells data from Benjamin J. Schmiedel et al., 2018 is available at https://dice-database.org/landing. The remaining data are available within the Article, Supplementary Information and Supplementary Data files. Source data are provided with this paper.

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

## Acknowledgements

We thank Biobank of the University Hospital Ostrava for providing access to patient samples, J.R. Bago for assistance with initiation of mouse experiments; J. Vrana for help with the flow cytometry experiments; and other members of the Blood Cancer Research Group for helpful suggestions and discussion. We also thank to Karel Harant and Pavel Talacko from Laboratory of Mass Spectrometry, Biocev, Charles University,

Faculty of Science, where proteomic and mass spectrometric analysis had been done. Cartoon in Fig. 8 was created with BioRender.com. This work was supported by The Czech Science Foundation GA CR 19-25354Y received by MS, GA CR 21-21413S received by MH, and by the New Directions of Biomedical Research in the Ostrava Region (No. CZ.02.1.01/0.0/0.0/18_069/0010060) finances provided from ERDF received by RH.

## Author contributions

Conceptualization, M.S., A.V., and T.J.; Methodology, A.V., M.S. M.D., and D.R.; Investigation, A.V., M.D., M.S., T.S., H.S., R.S., O.V. and M.H.; Visualization, M.T.; Formal analysis, T.S., D.Z., and T.P.; Resources, T.J., and R.H.; Writing – original draft, M.S., and A.V., Writing – review & editing, all authors; Funding acquisition, M.S., M.H., and R.H.; Supervision, M.S. and R.H.

## Competing interests

R.H. has had a consultant or advisory relationship with Janssen, Amgen, Celgene, AbbVie, BMS, Novartis, PharmaMar, and Takeda; has received honoraria from Janssen, Amgen, Celgene, BMS, PharmaMar, and Takeda; has received research funding from Janssen, Amgen, Celgene, BMS, Novartis, and Takeda. All additional authors declare no conflicts of interest.
