## [Peer Review File · Nature Communications]

The deubiquitinase OTUD1 regulates immunoglobulin production and proteasome inhibitor sensitivity in multiple myelomaREVIEWER COMMENTS

Reviewer #1 (Remarks to the Author):

The authors present a largely well-written and clearly presented manuscript in which they make the case for a role of the deubiquitinating enzyme, OTUD1, in myeloma cell Ig synthesis via PRDX4. The data are potentially interesting and pathophysiologically – and therefore clinically – relevant but require further confirmation. Moreover, the manuscript requires fine-tuning with improved referencing of relevant work in the field.

1. ABSTRACT: here, and throughout the manuscript, multiple definite and indefinite articles are missing. The manuscript should be reviewed by a native English speaker. Moreover, the abstract makes claims that require considerable further confirmation (detailed below).
2. INTRODUCTION: It is widely appreciated that serum Ig levels largely reflect tumour load (not production per cell). Generally, the authors should disentangle Ig synthesis/folding/secretion and both intracellular and extracellular/serum Ig levels more carefully.
3. the last paragraph (lines 47-56) lacks reference and discussion of important previous studies in normal and transformed PCs that are essential in this context. These include (but are not necessarily limited to) the following PMIDs: 16498407, 20651073, 22685320, 12374812.
4. RESULTS: the first two paragraphs are not very informative or novel (cf comment 2). The data underlying the paragraph around iIgL concentrations and prognosis (69-79) are weak, to demonstrate prognostic relevance the authors would have to use considerably more advanced multivariate analyses that take into account a range of molecular and clinical features (where they are simply dichotomising patient outcomes by high versus low iIgL levels).
5. Lines 98-99: the data shown are not strong enough to justify the statement that 'cumulative evidence...the importance of OTUD1 in plasma cell biology... particularly in Ig synthesis'. The authors need to show additional independent approaches, ideally a negative control (using overexpression and KD of another deubiquitinating enzyme). The authors also claim that OTUD12 is a 'tumour suppressor in MM' (paragraph title) based largely on slower tumour growth with OTUD1 overexpression and faster tumour growth with one shRNA in a subcutaneous xenograft model with one MM cell line (suboptimal in several ways). Additional cell lines and more shRNAs/other genetic approaches are needed to confirm this statement.
6. Line 138-139: 'Surprisingly, OTUD1 oe led to a massive rise in total ubiquitination (Fig. 3a) while 139 OTUD1 knock-down had the opposite effect (Fig. 3b).' This was only done with one shRNA and therefore needs further confirmation.
7. Line 139-141: 'As we observed this phenomenon only in the myeloma but no other cancer cell lines...'. Where do the authors show their results from other cell lines – I did not find those data. To support this claim, the authors should show at least one solid and one other haematopoietic (preferentially B cell) cancer cell line with at least 2 shRNAs or a comparable genetic method.
8. Line 145-146: please consider references mentioned in comment 3 and others.
9. Line 149-151: issues here are that only one shRNA was used and that PIs were used in a manner that is not mimicking in vivo exposure of MM cells (which is high levels of exposure of several hundred to thousand nM for 30-60min; the authors used low levels for 16h).
10. Line 151-152: 'Analysis of proteasome expression...not reveal any differences...' The authors only measure two subunits in one overexpression analysis, this is not sufficient to make the claim which would require testing several subunits for 19/20s each on mRNA or protein level.
11. Line 156: '...likely caused by overload of misfolded Ig molecules'. Increased Bodipy fluorescence does not allow this conclusion; where do the authors show an increase in MISFOLDED molecules/Ig? They do not show an increase in ER stress signalling either.
12. Line 232-234: can the authors clarify this – p5 shows very low levels of Ub, and correlating PFS in this simplistic manner is inappropriate (also see comment 4). I also suggest re-phrasing the conclusion in lines 236-237 as it is very strong and infers clonal processes in patients from one specific observation in cell lines.
13. Lines 241-250: the authors fail to show an effect on ER stress levels (which would require analysis of a set of key ER stress indicators such as HSPA5, DDIT3, P58IPK, PERK/eIF2alpha phosphorylation).
14. DISCUSSION, line 268: is OTUD1 a 'plasma cell specific enzyme' i.e. it is only expressed/active in PCs but not in any other cell type?
15. Line 275: in my view, the data are not showing a role for OTUD1 in Ig FOLDING – which data do the authors refer to?

16. The authors generally focus on Ig FOLDING issues but, as pointed above in comment 13, they show essentially no direct or indirect (ER stress markers) evidence of protein misfolding. They should also consider, and as a minimum DISCUSS, other stresses that may be relevant in relation to Ig folding and proteasomal capacity/inhibition such as oxidative stress and other metabolic challenges highlighted in relevant papers such as PMID 33883278, 23354484, 18241675.

17. FIGURES: authors need to show individual data points and reconsider if t-tests were used appropriately (has normal distribution of data been confirmed?). Can authors also show immunoblot replicates for the vast majority of IBs please.

Reviewer #2 (Remarks to the Author):

In this work, Vdovin and colleagues propose an unprecedented role played by the de-ubiquitinating enzyme, OTUD1, in the plasma cell (PC) cancer, multiple myeloma (MM). Their claims are novel and have potential therapeutic and prognostic value.

The story builds on the finding that the intracellular accumulation of the main PC secretory product, the immunoglobulin (Ig) light chain (iIgL) predicts better clinical outcome, a relevant observation per se. Being such accumulation not accounted for by high Ig transcripts, the authors posit that it indicates higher proteocatabolic burden on the ubiquitin (Ub) proteasome system (UPS), a central issue and a therapeutic target in MM pathobiology. Hence, they search for de-ubiquitinating enzymes (DUBs) essential for MM cell survival, and identify OTUD1, a known ovarian tumor suppressor. In further mechanistic efforts they reveal that OTUD1 controls the proteasome load, i.e., accumulation of poly-Ub proteins, and vulnerability to proteasome inhibitors (PI), fundamental anti-myeloma agents, possibly through a novel direct control exerted on Ig oxidative folding by interacting with, and increasing the abundance of, the endoplasmic reticulum (ER) resident oxidoreductase, peroxiredoxin 4 (PRDX4).

Overall, the research reported is sound, but the main conclusions drawn are not fully supported by the evidence provided. Several key results appear overlooked or not correctly interpreted, while others appear to conflict with the general model drawn. Thus, additional work is required, which may help the authors solidify the current conclusions, or perhaps even modify their model substantially.

The initial part on PC iIgL content as a prognostic marker is brilliant and of great clinical value; moreover, its exploitation as a biological framework to identify unprecedented mechanisms underlying PC protein homeostasis in search for potential new druggable targets appears powerful. Indeed, their identification of OTUD1 as a significant player in PC protein homeostasis seems solid. However, the paper is weak in the subsequent mechanistic dissection.

The main problems revolve around: 1) the mechanism and functional significance of the hypothesized association of OTUD1 with PRDX4; 2) the mechanism whereby PRDX4 mediates the effects of OTUD1 on protein homeostasis. These issues raise many unresolved questions, briefly summarized as follows:

- while PRDX4 appears causally involved (Figure 6), its association with OTUD1, a cytosolic DUB, is not convincingly demonstrated (Figure 5) and requires more experimental evidence;
- the mechanism whereby OTUD1 increases PRDX4 abundance (stabilization?) is not demonstrated;
- stabilization of PRDX4 by OTUD1 is assumed based on the DUB activity of the latter, but OTUD1 is cytosolic, and ubiquitination of a soluble ER resident protein within the ER prior to retro-translocation contradicts the current knowledge of ER-associated degradation (ERAD); moreover, DUBs are thought to downregulate retro-translocation of non-ubiquitinated ERAD substrates by regulating translocation machinery components, rather than by interacting with ERAD substrates, as in the case of OTUD2 (Christianson & Ye, Nat Struct Mol Biol. 2014). Is PRDX4 really a substrate of OTUD1? Is OTUD1 expressed in, or translocated to the ER lumen?
- OTUD1 increases iIgL protein levels without affecting transcript abundance and Ig secretion: the authors should determine the level at which iIgL protein expression is regulated. They should test if heightened iIgL stems from higher translation or reduced Ig folding capacity. The former possibility would offer an alternative regulation of PC secretory activity by OTUD1, also capable of burdening ERAD. Finally, the causal involvement of PRDX4 needs to be explained.
- If OTUD1 increases PRDX4 and empowers PRDX4-dependent Ig manufacture, this should alleviate, not aggravate, ERAD and proteasome workload: how can this be stressful to PCs? Since

ERAD disposes of terminally misfolded or inefficiently assembled ER client proteins, do the authors hypothesize that PRDX4 accumulating upon OTUD1 overexpression is dysfunctional and in fact disturbing the secretory pathway?

The resulting experimental requests, along with additional compulsory and minor issues are detailed as follows.

Major, compulsory issues

- Title: the title conveys the incorrect message that OTUD1 is a tumor suppressor in multiple myeloma. OTUD1 is a tumor suppressor in other cancers, and the proposed mechanism may underlie an oncosuppressive function, but a tumor suppressor should reveal inactivating mutations with a role in the genesis and progression of a given cancer. The authors should either remove "tumor suppressor" from the title (best option) or rephrase it: "The tumor suppressor OTUD1 mediates (...) in multiple myeloma".
- Figure 1. The results of the CRISPR screening for candidate essential DUBs should be shown as a supplementary item.
- A general concern throughout the results, in all protein biochemistry experiments the authors must show average quantifications of repeated independent experiments.
- Page 5, line 121 ("secreted IgL") and Supplemental Figure 2c. Excluding an effect of OTUD 1 on Ig secretion is key in the paper's context. The authors should formally measure Ig secretion in proper standard secretion assays.
- Page 6, line 153, and Suppl. Figure 3d. On "profound occupation" of the proteasome in OTUD1 overexpressing MM cells: 1) in the absence of a positive control (e.g. cells with proteasomes clogged by a poly-Q proteotoxin, or irreversibly inhibited) the extent of occupation cannot be determined; Fig S3d only suggests that proteasomes are less accessible to the dye upon OTUD1 overexpression; 2) the legend to that panel incorrectly refers to "proteasome capacity", which is measured in S3c, rather than "occupancy", as it should; 3) the methods of the assay shown in Fig.S3d are missing.
- Figure 5: the colocalization of PRDX4 and OTUD1 in the ER is not conclusively demonstrated. First, the co-immunoprecipitation of endogenous OTUD1 and PRDX4, a central issue, is not convincing. The authors should show significant co-IP of OTUD1 also with endogenous PRDX4. The requirement of H₂O₂ for association of recombinant PRDX4 with OTUD1 in vitro is not sufficient evidence for the actual association of the real proteins in the ER in cells. Subcellular ER and cytoplasmic localization should also be tested by standard fractionation experiments. Moreover, in view of their supposedly incompatible subcellular localization of PRDX4 and OTUD1, imaging-based evidence of their colocalization is needed. Suitable ER markers are also needed to demonstrate actual localization of OTUD1 to the ER.
- Figure 5h: the authors cannot conclude that OTUD1 protects PRDX4 from degradation, i.e. stabilizes it, in the absence of: 1) evidence that ERAD (proteasome) inhibitors prevent OTUD1 knockdown from reducing PRDX4 abundance; 2) cycloheximide (CHX) assays showing differential reduction over time of the PRDX4 pool when OTUD1 is overexpressed or silenced, following CHX administration.
- The authors should carefully determine the level at which IgL protein expression is regulated by OTUD1. First, they should convincingly exclude an effect on Ig secretion in proper secretion assays. Then, they should test if heightened iIgL in OTUD1 overexpressing cells stems from higher translation or reduced Ig folding capacity. Higher translation would offer an alternative mechanistic view of how OTUD1 may regulate PC secretory activity resulting in increased ERAD burden. Once clarified these issues, efforts should be made to explain the causal implication of PRDX4.
- Figure 6: although PRDX4 appears causally involved in the impact of OTUD1 on accumulation of iIgL and poly-Ub proteins and on PI sensitivity, it does not mediate the effect of OTUD1 on cell proliferation, convincingly demonstrated in vitro and in vivo (Figure 2). How do the authors explain this apparent discrepancy? This should be discussed carefully, rather than simply repeated in a line in the discussion (page 9, lines 272-3). In view of this, further caution should be used when proposing the identified circuit as a possible tumor suppressing mechanism. The authors should also test if the effect of OTUD1 on proliferation requires IgL expression.
- Figure 7. To prove their model, the authors should prove PRDX4 to be a direct substrate of OTUD1. Can they exclude a role of OTUD1 on the retrotranslocation machinery affecting ERAD and overall proteostasis (similar to that demonstrated for OTUD2 in Bernardi KM et al, Mol Biol Cell 2013; 24:3545-56)?

- Increased abundance of PRDX4 in the ER should boost Ig manufacture and alleviate, not aggravate, proteasome workload: how do the authors explain the stressful effects of OTUD1 in PCs? Is excess PRDX4 dysfunctional and in fact disturbing Ig folding and assembly?

Minor points

- Page 4, line 91: "covering most human DUBs" should better read "covering most currently annotated human DUBs". Is there an established number of DUBs? Are all DUBs known?
- Page 4, line 92: "alter Ig production" should better read "intracellular IgL accumulation".
- Page 5, line 118: "Similar to MM patients" is misleading. Best something like: "Suggesting a cause-effect relationship of the association documented in patients..."
- Page 6, lines 143-4: It is incorrect to conclude that "misfolded Ig are likely the majority of ubiquitinated proteins in OTUD1 or cells"; rather, they appear to be the major source of burden for the ubiquitin-proteasome system in MM cells.
- Page 6, lines 146-7: It is incorrect to conclude that "OTUD1 activity correlates with the amount of IgL ubiquitinated products", as the amount of Ub-IgL has not been determined. Say "the amount of ubiquitinated protein species" instead.
- Doxycycline has been shown to possess anti-aggregation activity (e.g., Giorgetti et al, J Biol Chem 2011 286(3):2121-31): the authors should control for the possible effect of doxycyclin at the dosage used to induce transgene expression on relevant proteostasis-related parameters (in primis, accumulation of poly-ubiquitinate proteins and PI toxicity).
- Figure 4: clearly specify the number of mice per experimental group in the in vivo studies. Are all mice, including controls, treated with doxycycline?
- Page 7, lines 174-181: 1) is OTUD1 really ubiquitinated by KEAP1? Fig. 4f hardly shows. Need replicates and quantifications of at least three independent expts; 2) if it is, since ETGE sequence removal fails to affect OTUD1 protein levels both basal and upon protein synthesis blockade (CHX), must explicitly conclude that KEAP1-dependent ubiquitination of OTUD1 does not control protein stability; 3) in view of the conclusions excluding KEAP1 from mediating the effects of OTUD1, is figure 4 of use?
- Page 7, lines 202-3: panel G must be cited after F in the text; shift the panels accordingly.
- Page 9, line 266-7. That Ig production is controlled transcriptionally is hard to dispute, please rephrase: "These findings indicate that the control exerted by OTUD1 on Ig relevant to MM proteostasis is regulated mainly at the post-transcriptional level".

Reviewer #3 (Remarks to the Author):

In this paper, Dr. Vdovin and colleagues evaluate the role of the deubiquitinating (DUB) enzyme OTUD1 in regulating the intracellular synthesis and folding of Ig (iIgL). They observe that levels of iIgL correlate with PFS of patients treated with bortezomib. By performing a DUB KD they identify OTUD1 as a regulator of iIgL production. They generated two models of OTUD1 OE and KD in MM1S and RPMI to infer its effect on cell growth, in vitro and in vivo. They demonstrate the impact of OTUD1 in the ubiquitin pools in MM cells. They further show that OTUD1 is regulated by the E3 ligase KEAP1 and regulates PRDX4, which participates to the ER redox cycles.

The study is well conducted and clearly exposed. The conclusions of the paper are supported by the experiments. I have only minor comments:

Fig. 1b. It would be important to provide information regarding those patients. As for survival analysis in Fig. 1c and i, are those patients treated homogeneously ?

Fig. 2. Is the overexpression of OTUD1 inducing a decreased cell proliferation or apoptosis?

Fig. 3c. WB of HA-OTUD-1 does not look very clear.

Fig. 6. Have you explore the ER pathway in your models of OTUD1-KD and OTUD1-OE, perhaps with KEAP1 rescue experiments ?

RESPONSE TO REVIEWERS' COMMENTS

Reviewer #1 (Remarks to the Author):

The authors present a largely well-written and clearly presented manuscript in which they make the case for a role of the deubiquitinating enzyme, OTUB1, in myeloma cell Ig synthesis via PRDX4. The data are potentially interesting and pathophysiologically – and therefore clinically - relevant but require further confirmation. Moreover, the manuscript requires fine-tuning with improved referencing of relevant work in the field.

We thank the reviewer for acknowledging the clinical potential of our study and particularly for very helpful and constructive comments. In responding to them, our manuscript has strongly improved. Our responses and additional analyses should clarify both the clinical and mechanistic aspects of the study, thereby further enhancing the manuscript.

1. ABSTRACT: here, and throughout the manuscript, multiple definite and indefinite articles are missing. The manuscript should be reviewed by a native English speaker. Moreover, the abstract makes claims that require considerable further confirmation (detailed below).

Many thanks for spotting the language issues. The re-submitted version of the manuscript has been proofread and corrected by a native English speaker. Similarly, we have adjusted the abstract text and title of the article according to the newly acquired data.

2. INTRODUCTION: It is widely appreciated that serum Ig levels largely reflect tumour load (not production per cell). Generally, the authors should disentangle Ig synthesis/folding/secretion and both intracellular and extracellular/serum Ig levels more carefully.

We changed the introduction text to define pathways and consequences of Ig production and its clinical relevancy more precisely.

3. the last paragraph (lines 47-56) lacks reference and discussion of important previous studies in normal and transformed PCs that are essential in this context. These include (but are not necessarily limited to) the following PMIDs: 16498407, 20651073, 22685320, 12374812.

Thank you for your inspirational suggestions. We apologize for omitting the referencing of several previous studies. Indeed, we used some of them when creating our hypothesis. We included the suggested references and found them highly relevant to our research.

4. RESULTS: the first two paragraphs are not very informative or novel (cf comment 2). The data underlying the paragraph around iLgL concentrations and prognosis (69-79) are weak, to demonstrate prognostic relevance the authors would have to use considerably more advanced multivariate analyses that take into account a range of molecular and clinical features (where they are simply dichotomising patient outcomes by high versus low iLgL levels).

To further support the conclusion drawn from Fig. 1b, d, we have performed a multivariate analysis of our patient cohort. The results confirmed the unique predictive potential of iLgL over other clinical and molecular parameters. We have also adjusted the introduction text accordingly.

5. Lines 98-99: the data shown are not strong enough to justify the statement that 'cumulative evidence...the importance of OTUD1 in plasma cell biology... particularly in Ig synthesis'. The authors need to show additional independent approaches, ideally a negative control (using overexpression and KD of another deubiquitinating enzyme). The authors also claim that OTUD12 is a 'tumour suppressor in MM' (paragraph title) based largely on slower tumour growth with OTUD1 overexpression and faster tumour growth with one shRNA in a subcutaneous xenograft model with one MM cell line (suboptimal in several ways). Additional cell lines and more shRNAs/other genetic approaches are needed to confirm this statement.

1) We performed new experiments using knock-downs of 6 other deubiquitinating enzymes in MM cells. These data are presented in the new Supplementary Fig. 2g.

2) We removed statements about OTUD1 being a tumour suppressor in MM both from the title and throughout the manuscript's text.

3) We added new results using second shRNA targeting OTUD1 to Fig. 2g. Additionally, we provide new Supplementary Fig. 2l that further disentangles role of OTUD1 in IgL synthesis and plasma cell proliferation.

6. Line 138-139: 'Surprisingly, OTUD1 oe led to a massive rise in total ubiquitination (Fig. 3a) while 139 OTUD1 knock-down had the opposite effect (Fig. 3b).' This was only done with one shRNA and therefore needs further confirmation.

We included results using second shRNA targeting OTUD1 to Fig. 3b.

7. Line 139-141: 'As we observed this phenomenon only in the myeloma but no other cancer cell lines...'. Where do the authors show their results from other cell lines – I did not find those data. To support this claim, the authors should show at least one solid and one other haematopoietic (preferentially B cell) cancer cell line with at least 2 shRNAs or a comparable genetic method.

We added additional experiments using non-hematopoietic cell line (HEK293) and non-plasma cell B-cell line (Raji). These results are presented in Supplementary Fig. 3a.

8. Line 145-146: please consider references mentioned in comment 3 and others.

We added the suggested references.

9. Line 149-151: issues here are that only one shRNA was used and that PIs were used in a manner that is not mimicking in vivo exposure of MM cells (which is high levels of exposure of several hundred to thousand nM for 30-60min; the authors used low levels for 16h).

We included additional results using the second shRNA in Fig. 3g,h. Clinical pharmacokinetics studies have shown that after intravenous injection of bortezomib at the recommended dose (1,3 mg/m²), it reaches the maximum concentration in plasma 112ng/ml (291nM), and the elimination half-life is more than 40h (PMID: 29802543). Cell line growing in cultural media is a rather simplistic model, which does not mimic many crucial parameters affecting MM cells in patients, such as stromal cell surrounding and intracellular matrix, the concentration of O₂ and CO₂, drug metabolism by the liver, etc. Therefore, it is hard to directly compare concentrations of a drug in the patient's bone marrow and culture media (PMID: 22825471). Our goal was to analyse the effect of OTUD1 expression on drug sensitivity using MM cell lines (RPMI8226 and MM1.S) as model objects; therefore, we had chosen timing and concentration in order to achieve reasonable viability under experimental conditions. Low concentrations of bortezomib with longer exposure have been used for RPMI8226 and MM1.S in multiple studies (e.g., PMID: 33250513, 22825471, 23531700).

10. Line 151-152: 'Analysis of proteasome expression...not reveal any differences...' The authors only measure two subunits in one overexpression analysis, this is not sufficient to make the claim which would require testing several subunits for 19/20s each on mRNA or protein level.

We extended the analysis presented in Supplementary Fig. 4b,c using additional proteasome subunits.

11. Line 156: '...likely caused by overload of misfolded Ig molecules'. Increased Bodipy fluorescence does not allow this conclusion; where do the authors show an increase in MISFOLDED molecules/Ig? They do not show an increase in ER stress signalling either.

- 1) We extended the Bodipy analysis using OTUD1 oe MM cells transfected with silgL (Fig. 3f) to show that the increase of IgL is responsible for higher occupation of proteasome in OTUD1 oe.
- 2) Results presented in Fig. 3c indicate that significant pool of ubiquitinated proteins in MM cells are IgL molecules.

- 3) We included new Fig. 3d,e showing differences in IgL ubiquitination in MM cells with altered OTUD1 expression. Because IgL can be ubiquitinated exclusively during its retrotranslocation from ER to cytosol, we concluded that OTUD1 controls the production of IgL by affecting its folding state.
- 4) We provide an analysis of p-eIF2a, a global marker of ER stress. The phosphorylation status of eIF2a positively correlates with OTUD1 expression, IgL levels and ubiquitination pattern (Fig. 3a-c, Fig. 7d).

12. Line 232-234: can the authors clarify this – p5 shows very low levels of Ub, and correlating PFS in this simplistic manner is inappropriate (also see comment 4). I also suggest re-phrasing the conclusion in lines 236-237 as it is very strong and infers clonal processes in patients from one specific observation in cell lines.

We adjusted our conclusions based on Fig. 7a-d accordingly.

13. Lines 241-250: the authors fail to show an effect on ER stress levels (which would require analysis of a set of key ER stress indicators such as HSPA5, DDIT3, P58IPK, PERK/eIF2alpha phosphorylation).

We added a global ER stress marker staining, p-eIF2a, to Fig. 3a-c and Fig. 7d.

14. DISCUSSION, line 268: is OTUD1 a 'plasma cell specific enzyme' i.e. it is only expressed/active in PCs but not in any other cell type?

OTUD1 is also expressed in other cell types and tissues. Therefore, we changed the wording accordingly.

15. Line 275: in my view, the data are not showing a role for OTUD1 in Ig FOLDING – which data do the authors refer to?

Based on Fig. 3c and new results presented in Fig. 3d-f and Fig. 6i we conclude that a significant pool of ubiquitinated proteins in MM cells consists of ubiquitinated IgL. Ubiquitination of IgL can occur only during ERAD, when IgL is re-translocated from the ER lumen to cytosol and destined to the proteasome. Therefore, we believe that the ubiquitinated forms of IgL represent the pool of misfolded IgL.

16. The authors generally focus on Ig FOLDING issues but, as pointed above in comment 13, they show essentially no direct or indirect (ER stress markers) evidence of protein misfolding. They should also consider, and as a minimum DISCUSS, other stresses that may be relevant in relation to Ig folding and proteasomal capacity/inhibition such as oxidative stress and other metabolic challenges highlighted in relevant papers such as PMID 33883278, 23354484, 18241675.

As also specified above, we addressed IgL folding in several new Figure panels (Fig. 3d-f and Fig. 6i, j). Additionally, we extended Fig. 3a-c with an analysis of the general ER stress marker p-eIF2a. Accordingly, we adjusted the related text in the results section. Similarly, we extended the discussion to describe the additional factors that might affect Ig folding status and proteasome capacity more precisely.

17. FIGURES: authors need to show individual data points and reconsider if t-tests were used appropriately (has normal distribution of data been confirmed?). Can authors also show immunoblot replicates for the vast majority of IBs please.

We re-analyzed the significance of results using the nonparametric Mann–Whitney U test in cases where we could not confirm the normal data distribution. Additionally, we changed the graphs showing the individual data points, where ever applicable (Fig. 1g,h and Fig. 2l,m). We provide both original uncropped versions of all immunoblots and their replicates in the Source Data file.

Reviewer #2 (Remarks to the Author):

In this work, Vdovin and colleagues propose an unprecedented role played by the de-ubiquitinating enzyme, OTUD1, in the plasma cell (PC) cancer, multiple myeloma (MM). Their claims are novel and have potential therapeutic and prognostic value.

The story builds on the finding that the intracellular accumulation of the main PC secretory product, the immunoglobulin (Ig) light chain (ilgL) predicts better clinical outcome, a relevant observation per se. Being such accumulation not accounted for by high Ig transcripts, the authors posit that it indicates higher proteocatabolic burden on the ubiquitin (Ub) proteasome system (UPS), a central issue and a therapeutic target in MM pathobiology. Hence, they search for de-ubiquitinating enzymes (DUBs) essential for MM cell survival, and identify OTUD1, a known ovarian tumor suppressor. In further mechanistic efforts they reveal that OTUD1 controls the proteasome load, i.e., accumulation of poly-Ub proteins, and vulnerability to proteasome inhibitors (PI), fundamental anti-myeloma agents, possibly through a novel direct control exerted on Ig oxidative folding by interacting with, and increasing the abundance of, the endoplasmic reticulum (ER) resident oxidoreductase, peroxiredoxin 4 (PRDX4).

Overall, the research reported is sound, but the main conclusions drawn are not fully supported by the evidence provided. Several key results appear overlooked or not correctly interpreted, while others appear to conflict with the general model drawn. Thus, additional work is required, which may help the authors solidify the current conclusions, or perhaps even modify their model substantially.

The initial part on PC ilgL content as a prognostic marker is brilliant and of great clinical value; moreover, its exploitation as a biological framework to identify

unprecedented mechanisms underlying PC protein homeostasis in search for potential new druggable targets appears powerful. Indeed, their identification of OTUD1 as a significant player in PC protein homeostasis seems solid. However, the paper is weak in the subsequent mechanistic dissection.

We are grateful to the reviewer for acknowledging our study's prognostic potential and clinical value. We agree that the mechanistic part of our manuscript contained several significant gaps which we believe to fill in the revised version. Based on the newly acquired data, we have strongly improved especially the mechanistic section of our work. Please see our point-by-point response below (major issues / minor points).

The main problems revolve around:

- 1) the mechanism and functional significance of the hypothesized association of OTUD1 with PRDX4;
- 2) the mechanism whereby PRDX4 mediates the effects of OTUD1 on protein homeostasis.

These issues raise many unresolved questions, briefly summarized as follows:

Responses and new experimental data related to the all questions below are provided in the following sections (Major, compulsory issues and Minor points).

- while PRDX4 appears causally involved (Figure 6), its association with OTUD1, a cytosolic DUB, is not convincingly demonstrated (Figure 5) and requires more experimental evidence;
- the mechanism whereby OTUD1 increases PRDX4 abundance (stabilization?) is not demonstrated;
- stabilization of PRDX4 by OTUD1 is assumed based on the DUB activity of the latter, but OTUD1 is cytosolic, and ubiquitination of a soluble ER resident protein within the ER prior to retro-translocation contradicts the current knowledge of ER-associated degradation (ERAD); moreover, DUBs are thought to downregulate retro-translocation of non-ubiquitinated ERAD substrates by regulating translocation machinery components, rather than by interacting with ERAD substrates, as in the case of OTUD2 (Christianson & Ye, Nat Struct Mol Biol. 2014). Is PRDX4 really a substrate of OTUD1? Is OTUD1 expressed in, or translocated to the ER lumen?
- OTUD1 increases IgL protein levels without affecting transcript abundance and Ig secretion: the authors should determine the level at which IgL protein expression is regulated. They should test if heightened IgL stems from higher translation or reduced Ig folding capacity. The former possibility would offer an alternative

regulation of PC secretory activity by OTUD1, also capable of burdening ERAD. Finally, the causal involvement of PRDX4 needs to be explained.

- If OTUD1 increases PRDX4 and empowers PRDX4-dependent Ig manufacture, this should alleviate, not aggravate, ERAD and proteasome workload: how can this be stressful to PCs? Since ERAD disposes of terminally misfolded or inefficiently assembled ER client proteins, do the authors hypothesize that PRDX4 accumulating upon OTUD1 overexpression is dysfunctional and in fact disturbing the secretory pathway?

The resulting experimental requests, along with additional compulsory and minor issues are detailed as follows.

Major, compulsory issues

- Title: the title conveys the incorrect message that OTUD1 is a tumor suppressor in multiple myeloma. OTUD1 is a tumor suppressor in other cancers, and the proposed mechanism may underlie an oncosuppressive function, but a tumor suppressor should reveal inactivating mutations with a role in the genesis and progression of a given cancer. The authors should either remove “tumor suppressor” from the title (best option) or rephrase it: “The tumor suppressor OTUD1 mediates (...) in multiple myeloma”.

We agree that the potential oncosuppressive role of OTUD1 in multiple myeloma is likely not dependent of its function in IgL production. This is supported by new Supplementary Fig. 2I. Therefore, we changed the title and removed/rephrased statements about the tumour-suppressive role of OTUD1 throughout the manuscript.

- Figure 1. The results of the CRISPR screening for candidate essential DUBs should be shown as a supplementary item.

We moved this Figure panel to Supplementary Fig. 1d.

- A general concern throughout the results, in all protein biochemistry experiments the authors must show average quantifications of repeated independent experiments.

We provide the Source Data file that contains repeats of the biochemistry experiments, including uncropped western blot images and raw data from additional analysis.

- Page 5, line 121 (“secreted IgL”) and Supplemental Figure 2c. Excluding an effect of OTUD 1 on Ig secretion is key in the paper’s context. The authors should formally measure Ig secretion in proper standard secretion assays.

We agree that excluding the potential effect of OTUD1 on the secretory pathway is an absolute necessity. We obtained the value of secreted and intracellular IgL presented in Supplementary Fig. 2c always from the single cell culture. OTUD1 oe leads to an increase in both intracellular and secreted IgL, while in the case of sh OTUD1 concentration of both secreted and intracellular IgL dropped. Any effect of OTUD1 on the IgL secretory pathway would lead to changes in the ratio of secreted vs. intracellular IgL, which we did not observe. Secretion of Ig in multiple myeloma cell lines is generally assessed by specific ELISA assays (see, e.g., doi.org/10.1182/blood.2019004147 Figure 5 H-I). To the best of our knowledge, we are not aware of any other type of secretory assay.

- Page 6, line 153, and Suppl. Figure 3d. On “profound occupation” of the proteasome in OTUD1 overexpressing MM cells: 1) in the absence of a positive control (e.g. cells with proteasomes clogged by a poly-Q proteotoxin, or irreversibly inhibited) the extent of occupation cannot be determined; Fig S3d only suggests that proteasomes are less accessible to the dye upon OTUD1 overexpression; 2) the legend to that panel incorrectly refers to “proteasome capacity”, which is measured in S3c, rather than “occupancy”, as it should; 3) the methods of the assay shown in Fig.S3d are missing.

1) To more precisely assess the level of proteasome capacity in cells with OTUD1 oe, we extended the Bodipy analysis using MM cells treated with carfilzomib, an irreversible proteasome inhibitor (now in new Fig. 3f).

2) Additionally, we knocked down IgL in OTUD1 oe cells and showed that increase in IgL is responsible for higher proteasome occupation (Fig. 3f).

3) We changed the legend of Fig. 3f as suggested.

- Figure 5: the colocalization of PRDX4 and OTUD1 in the ER is not conclusively demonstrated. First, the co-immunoprecipitation of endogenous OTUD1 and PRDX4, a central issue, is not convincing. The authors should show significant co-IP of OTUD1 also with endogenous PRDX4. The requirement of H₂O₂ for association of recombinant PRDX4 with OTUD1 in vitro is not sufficient evidence for the actual association of the real proteins in the ER in cells. Subcellular ER and cytoplasmic localization should also be tested by standard fractionation experiments. Moreover, in view of their supposedly incompatible subcellular localization of PRDX4 and OTUD1, imaging-based evidence of their colocalization is needed. Suitable ER markers are also needed to demonstrate actual localization of OTUD1 to the ER.

1) We added new results from co-immunoprecipitation of endogenous OTUD1 and PRDX4 (Fig. 5b) in MM cells.

2) We removed the original Fig. 5d (interaction of recombinant OTUD1 and PRDX4 in the presence of H₂O₂), changed the manuscript text, and presented the model accordingly (Fig. 7e).

3) We performed a subcellular fractionation experiment to separate ER and cytosol using full-length PRDX4 and signal sequence-deficient (delta1-38) PRDX4. (Supplementary Fig. 5c,d)

4) We analysed the subcellular localization of OTUD1 and PRDX4 using immunofluorescence (Fig. 5d,e).

- Figure 5h: the authors cannot conclude that OTUD1 protects PRDX4 from degradation, i.e. stabilizes it, in the absence of: 1) evidence that ERAD (proteasome) inhibitors prevent OTUD1 knockdown from reducing PRDX4 abundance; 2) cycloheximide (CHX) assays showing differential reduction over time of the PRDX4 pool when OTUD1 is overexpressed or silenced, following CHX administration.

1) We included new experimental data supporting the protective role of OTUD1 and PRDX4 (full length vs. delta1-38) using MG132 treatment (Fig. 5m,n).

2) We added new results obtained from OTUD1 oe cells treated with cycloheximide (Fig. 5k).

- The authors should carefully determine the level at which IgL protein expression is regulated by OTUD1. First, they should convincingly exclude an effect on Ig secretion in proper secretion assays. Then, they should test if heightened ilgL in OTUD1 overexpressing cells stems from higher translation or reduced Ig folding capacity. Higher translation would offer an alternative mechanistic view of how OTUD1 may regulate PC secretory activity resulting in increased ERAD burden. Once clarified these issues, efforts should be made to explain the causal implication of PRDX4.

1) We believe that results presented in Supplementary Fig. 3c exclude the potential role of OTUD1 in IgL secretory pathway. See also our explanation/response above.

2) We assessed the effect of OTUD1 on the rate of global protein synthesis and specific translation of IgL using the SUnSET puromycin assay (Supplementary Fig. 2d,e).

3) To further clarify the implication of PRDX4 in IgL synthesis and folding, we analysed changes in the overall ubiquitin pool and specifically ubiquitination of IgL (Fig. 6i,j). Presented results suggest that levels of PRDX4 directly correlate with IgL ubiquitination that indicates IgL folding state – only retrotranslocated (misfolded) IgL can undergo ubiquitination.

- Figure 6: although PRDX4 appears causally involved in the impact of OTUD1 on accumulation of IgL and poly-Ub proteins and on PI sensitivity, it does not mediate the effect of OTUD1 on cell proliferation, convincingly demonstrated in vitro and in vivo (Figure 2). How do the authors explain this apparent discrepancy? This should be discussed carefully, rather than simply repeated in a line in the discussion (page 9, lines 272-3). In view of this, further caution should be used when proposing the identified circuit as a possible tumor suppressing mechanism. The authors should also test if the effect of OTUD1 on proliferation requires IgL expression.

The new experimental analysis presented in Supplementary Fig. 2I indicates that the effect of OTUD1 on MM cell proliferation does not require IgL expression. On the other hand, OTUD1-mediated increase in IgL levels depends entirely on PRDX4 (Fig. 6c,d). In line with that, we extended the related parts of the results description and discussion in the manuscript. Additionally, we adjusted statements positioning OTUD1 to a tumour suppressor.

- Figure 7. To prove their model, the authors should prove PRDX4 to be a direct substrate of OTUD1. Can they exclude a role of OTUD1 on the retrotranslocation machinery affecting ERAD and overall proteostasis (similar to that demonstrated for OTUD2 in Bernardi KM et al, Mol Biol Cell 2013; 24:3545-56)?

We added new results showing the levels of protein disulfide isomerase (PDIA6), another ER lumen protein, are not affected by OTUD1 (Fig. 5m). Additionally, we show that OTUD1 specifically deubiquitinates the full length, ER-localised PRDX4 but not the cytosolic, delta1-38 PRDX4 (Fig. 5l).

- Increased abundance of PRDX4 in the ER should boost Ig manufacture and alleviate, not aggravate, proteasome workload: how do the authors explain the stressful effects of OTUD1 in PCs? Is excess PRDX4 dysfunctional and in fact disturbing Ig folding and assembly?

The limiting step in Ig folding is the reduction of oxidized PDI and H₂O₂ mediated primarily by PRDX4. In our model, we propose that increased levels of PRDX4 potentiate IgL production by participating in the oxidoreductive circuits involving the PDI-ERO1-PRDX4 axis. The selective stabilization of PRDX4 accelerates PDI-mediated IgL folding without directly affecting the ratio of correctly folded and misfolded IgL. Therefore, the elevated IgL production is accompanied by an increase in both folded and misfolded IgL. The latter ultimately saturates the capacity of available proteasomes and increases the ER stress.

Minor points

- Page 4, line 91: “covering most human DUBs” should better read “covering most currently annotated human DUBs”. Is there an established number of DUBs? Are all DUBs known?

We changed the wording according to the suggestion. Currently, about 100 DUBs have been annotated in the human genome (PMID: 28959952); however, we can rule out the possibility that future research will provide additional members of this enzymatic family.

- Page 4, line 92: “alter Ig production” should better read “intracellular IgL accumulation”.

We have changed the wording according to the suggestion.

- Page 5, line 118: “Similar to MM patients” is misleading. Best something like: “Suggesting a cause-effect relationship of the association documented in patients...”

We have changed the wording according to the suggestion.

- Page 6, lines 143-4: It is incorrect to conclude that “misfolded Ig are likely the majority of ubiquitinated proteins in OTUD1 or cells”; rather, they appear to be the major source of burden for the ubiquitin-proteasome system in MM cells.

We have changed the wording according to the suggestion.

- Page 6, lines 146-7: It is incorrect to conclude that “OTUD1 activity correlates with the amount of IgL ubiquitinated products”, as the amount of Ub-IgL has not been determined. Say “the amount of ubiquitinated protein species” instead.

We have performed a new set of experiments (Fig. 3d, e) that support our conclusion.

- Doxycycline has been shown to possess anti-aggregation activity (e.g., Giorgetti et al, J Biol Chem 2011 286(3):2121-31): the authors should control for the possible effect of doxycyclin at the dosage used to induce transgene expression on relevant proteostasis-related parameters (in primis, accumulation of poly-ubiquitinated proteins and PI toxicity).

The revisited version of the manuscript includes additional data supporting the role of OTUD1 and Ub-IgL in the saturation of proteasome. We performed the requested analysis to determine whether doxycycline treatment induces changes in the pool of poly-Ub proteins. The results are shown here:

- Figure 4: clearly specify the number of mice per experimental group in the in vivo studies. Are all mice, including controls, treated with doxycycline?

We added data points for each animal to the graph in Fig. 2l, m. Both control and OTUD1 oe mice were injected with the same MM cell line. To induce OTUD1 overexpression, we administered doxycycline only to the OTUD1 oe group. Our previous experimental work has proven no effect of the doxycycline (identical dosage and timing) on the xenograft growth.

- Page 7, lines 174-181: 1) is OTUD1 really ubiquitinated by KEAP1? Fig. 4f hardly shows. Need replicates and quantifications of at least three independent expts; 2) if it is, since ETGE sequence removal fails to affect OTUD1 protein levels both basal and upon protein synthesis blockade (CHX), must explicitly conclude that KEAP1-dependent ubiquitination of OTUD1 does not control protein stability; 3) in view of the conclusions excluding KEAP1 from mediating the effects of OTUD1, is figure 4 of use?

1) We performed new pull-down experiments that are now presented in Fig. 4f.

2) We changed the statement to explicitly conclude that KEAP1-mediated ubiquitination of OTUD1 does not lead to its degradation.

3) Our proximity labelling analysis repeatedly identified KEAP1 as one of the top OTUD1 interaction partners. We find it absorbing that OTUD1 participates in the Ig production that is inherently linked to the generation of oxygen radicals. At the same time OTUD1 directly binds KEAP1, a crucial sensor of cellular oxidative stress. Lack of phenotypes using OTUD1 deltaETGE (KEAP1 binding mutant) might suggest the potential existence of a parallel, yet to be identified regulatory pathway. Moreover, in another cellular (non-plasma cell) system, the OTUD1-KEAP1 complex might play a significant role, and thus we find our data highly inspirational for other researchers.

- Page 7, lines 202-3: panel G must be cited after F in the text; shift the panels accordingly.

We changed the order of the Figure referencing accordingly.

- Page 9, line 266-7. That Ig production is controlled transcriptionally is hard to dispute, please rephrase: "These findings indicate that the control exerted by OTUD1 on Ig relevant to MM proteostasis is regulated mainly at the post-transcriptional level".

We changed the wording accordingly.

Reviewer #3 (Remarks to the Author):

In this paper, Dr. Vdovin and colleagues evaluate the role of the deubiquitinating (DUB) enzyme OTUD1 in regulating the intracellular synthesis and folding of Ig (IgL). They observe that levels of IgL correlate with PFS of patients treated with bortezomib. By performing a DUB KD they identify OTUD1 as a regulator of IgL production. They generated two models of OTUD1 OE and KD in MM1S and RPMI to infer its effect on cell growth, in vitro and in vivo. They demonstrate the impact of OTUD1 in the ubiquitin pools in MM cells. They further show that OTUD1 is regulated by the E3 ligase KEAP1 and regulates PRDX4, which participates to the ER redox cycles.

The study is well conducted and clearly exposed. The conclusions of the paper are supported by the experiments. I have only minor comments:

We thank the reviewer for acknowledging our study design and findings and for insightful suggestions to improve the experimental rigor of our manuscript.

Fig. 1b. It would be important to provide information regarding those patients. As for survival analysis in Fig. 1c and i, are those patients treated homogeneously ?

Information about the patient cohort used to generate Fig. 1c, d and Supplementary Fig. 1c are presented in the Supplementary Table 3. To exclude the potential effect of additional factors than IgL on the predictive potential, we have performed sophisticated multivariate analysis (Supplementary Table 4) that further strengthens the unique predictive potential of IgL concentration over other clinical and biochemical parameters.

Fig. 2. Is the overexpression of OTUD1 inducing a decreased cell proliferation or apoptosis?

To delineate the effect of OTUD1 on cell proliferation, we performed a viability test and cell cycle analysis. The results are now included in Supplementary Fig. 2j, k. OTUD1 oe did not change cell viability but led to an arrest in S phase of the cell cycle, which is reflected by slower proliferation.

Fig. 3c. WB of HA-OTUD-1 does not look very clear.

We replaced Fig. 3c with a new version.

Fig. 6. Have you explore the ER pathway in your models of OTUD1-KD and OTUD1-OE, perhaps with KEAP1 rescue experiments?

The revised manuscript contains staining of p-eIF2a, a general ER stress marker, in Fig. 3a-c and Fig. 7d. Additionally, based on subcellular fractionation and localization analysis (Fig. 5 and Supplementary Fig. 5c, d) we adjusted our mechanistic model of how OTUD1 stabilizes PRDX4.

REVIEWER COMMENTS

Reviewer #1 (Remarks to the Author):

The authors have responded adequately to a good number of comments. However, some issues remain concerning.

The first set of results, showing no correlation between secreted Ig and outcomes, is still weak as it is not novel. I still do not understand why the authors suggest that Ig secretion (i.e. serum paraprotein levels) would be widely considered as prognostic in any way, this simply is not the case.

The multivariate analysis of iIgL and responses to therapy (page 5, line 115-117): this appears to be based on 24 patients, 12 each in the intracellular Ig Low/high groups. It is not feasible to carry out a meaningful MV analysis with such low patient numbers, as every trial statistician will be able to confirm. Moreover, the CI is sizeable, and only three factors (why these) went into the MV analysis. The data are therefore not convincing. The authors should carry out a proper MV analysis, consider removing the clinical correlation attempts, or find another independent confirmation of their results.

Page 7, line 202-203, around IgL knockdown in OTUD1 oe cells (Fig 3c): the authors state that "importantly, the observed changes...directly correlated with the ER stress status". Where do they show 'ER stress status'? This is unclear to me. If they mean eIF2a phosphorylation, they need to be clear. eIF2a phosphorylation is also NOT a 'global ER stress marker' as the authors claim in their response to my comment 13, where I requested the analysis of a set of ER stress indicators. eIF2a phosphorylation occurs in response to an array of stresses and is carried out by four EIF2AKs. Unless the authors show increased expression on mRNA and/or protein level of adequate ER stress indicators such as chaperones HSPA5/P58IPK, and show PERK (EIF2AK3) phosphorylation, they cannot make any claims of 'ER stress'. This is a major point that should be addressed. In this context, the authors also need to explore the status of GCN2 (EIF2AK4) and potentially HRI. Both PERK and GCN2, and to some extent HRI, can also be functionally explored with highly selective inhibitors. In particular, GCN2 has recently been identified as a myeloma vulnerability in relation to proteasome inhibition and MYC signalling (PMID: 33883278; PMID: 34732728). Given that eIF2a phosphorylation is such an important marker here, it is critical the authors conduct some experiments that define if GCN2 or PERK are responsible for eIF2a phosphorylation in their experiments. Does PERK or GCN2 inhibition affect any of the downstream effects of eIF2a signalling? One easy and sensitive way is to look for effects on DDIT3 and HSPA5 expression by qRT-PCR/Western. Can the authors also show total eIF2a as a loading control please as it may be cleaved in stressed cells. GCN2 can also be activated by stalling ribosomes, which could be linked to deregulated translation control, so it is worth at least a few experiments as outlined above and a discussion of the roles of GCN2 in MM (PMID: 33883278; PMID: 34732728).

Response to comment 17 – individual data points: there remains an inexplicably large number of figures in which the authors chose not to show individual data points, when it is clear that they can and should be shown – which is essentially every bar chart. Also, can the authors please confirm in every relevant figure that results are from biologically independent replicates (as opposed to technical replicates) – this is particularly relevant for Fig 2f/g, Fig 3g/h, Fig 6i, Fig 7c. Also, have results shown in Fig. 4g and 5k been repeated?

Sensitisation to proteasome inhibition: I continue to maintain that short high-dose pulses best reflect PI exposure in vivo in patients, but as there is not agreed consensus for experimental approaches this cannot be resolved objectively, and I am therefore willing to accept their response. However, I urge them to rethink this approach in future work. The fact that others have used low concentrations for longer periods does not make a flawed approach right.

In this context, another issue is mechanistically troubling, which is the observation that expression of proteasome subunits did not change in OTUD1 oe cells (Suppl Fig 3b-d). If there was indeed a higher load on the proteasome, and ER stress with a UPR, one of the very first responses would be the transcriptional upregulation of proteasomes. How can the authors explain this discrepancy?

Reviewer #2 (Remarks to the Author):

The authors have addressed most of the issues raised. As a result, several passages have been clarified, and the manuscript grew substantially. However, few concerns have not been understood and still need to be addressed. But more importantly, the story still appears confusing, with its central issue, the link between OTUD1 and immunoglobulin production, being convincingly relevant, but mechanistically unresolved.

In general, the manuscript proves that OTUD1 controls secretory homeostasis and proteasome inhibitor (PI) sensitivity in myeloma cells through a mechanism mediated by the ER-resident peroxiredoxin, PRDX4. The authors also document an anti-proliferative effect of OTUD1, consistent with its proposed oncosuppressive role in other cancers, but this function appears separated from its myeloma-specific role in secretory homeostasis.

The paper aims to understand the latter; however, despite remarkable experimental efforts reported in the revised version, the story is still unclear. In a nutshell, OTUD1 increases intracellular Ig in myeloma cells, suggesting that it causes intracellular retention of Ig molecules, perhaps owing to defective Ig maturation along the secretory pathway, but this is in contrast with data that suggest that OTUD1 does not affect Ig mRNA expression, translation, and Ig secretion. The careful reader is left with the doubt as to whether the effect of OTUD1 on Ig folding should be investigated more carefully.

The authors also document a specific role of OTUD1 mediating deubiquitination of PRDX4, leading to its protection from ERAD and stabilization, and prove that PRDX4 is causally involved in the mechanism whereby OTUD1 increases intracellular Ig. They conclude that OTUD1 augments Ig production by stabilizing PRDX4, leading to UPS overload and enhanced PI sensitivity. The involvement of PRDX4 is convincing, but its effect on Ig retention and ERAD overload is counterintuitive and unexplained.

Although a novel role of an OTUD1-PRDX4 axis in PI sensitivity is a relevant and worth reporting finding, since the title and abstract (and entire manuscript) claim that "OTUD1 mediates Ig production", more conclusive and univocal mechanistic dissection, and thus additional, focused experimental work, in addition to very careful critical discussion, is expected.

Additional compulsory issues are detailed as follows.

Specific points.

1. Page 7: the title ("OTUD1 modulates IgL folding") is misleading, as Ig folding has not been directly investigated in this work. Please rephrase.
2. Figure 5: stabilization of PRDX4 by OTUD1 is key in the proposed role played by OTUD 1 in secretory proteostasis. To convincingly prove it, the Authors performed CHX assays, as requested, but the evidence generated completely relies on exogenously expressed proteins. A CHX experiment on endogenous PRDX4 is not only needed, but perfectly feasible in the Authors' hands, as they did measure endogenous PRDX4 abundance upon OTUD1 overexpression (Fig.5i). Essentially, they should simply add CHX treatment to the experiment shown in Fig.5i.
3. Few requests were misinterpreted and not addressed; they should be dealt with.
 - In Figure 1, I requested that the results of the CRISPR screening for candidate essential DUBs be displayed as a supplemental item. The authors misinterpreted my request, as they thought to address it by moving a scheme illustrating the strategy (a panel in original Figure 1) to supplemental Figure 1. The scheme was perfectly fine in its original location. Rather, I had requested to "display the results of the screening", i.e., the list of the identified DUBs, e.g. as a supplemental table.
 - The next request was to "show average quantifications of repeated independent experiments" in protein biochemistry experiments. The Authors respond that they provided original data. That is a laudable action, but again, my request was different. Please do what requested: in all protein biochemistry experiments, in addition to showing representative immunoblots, average

quantifications must be provided of the independent experiments run, and the number of independent experiments conducted specified in the legends.

4. Minor issues:

- Abstract, "the combinatory approach": is unclear.
- Page 10, 10th last line: OTUD1 failed to immunoprecipitate the PRDX4 mutant – not the other way round.
- Page 10, 2nd last line: Fig 5e is in fact 5f.

Reviewer #3 (Remarks to the Author):

The authors have answered all comments properly. I have no further comments.

RESPONSE TO REVIEWERS' COMMENTS

Reviewer #1 (Remarks to the Author):

The authors have responded adequately to a good number of comments. However, some issues remain concerning.

The first set of results, showing no correlation between secreted Ig and outcomes, is still weak as it is not novel. I still do not understand why the authors suggest that Ig secretion (i.e. serum paraprotein levels) would be widely considered as prognostic in any way, this simply is not the case.

To the best of our knowledge, the presented analysis of M-protein levels (secreted Ig, serum paraprotein) vs patient outcome (PFS, OS) represents the largest cohort ever described in the literature (>4000 patients). We do not refer serum M-protein to be a prognostic but rather an important and a common diagnostic marker in nearly all patients with monoclonal gammopathies, particularly multiple myeloma. The correlation analysis is shown as a contrast to the recent report (PMID: 30833275) that suggested a prognostic effect of M-protein levels in newly diagnosed myeloma patients.

To clarify our conclusions, in the revised version of the manuscript, we are again clearly stating that serum M-protein has no prognostic impact in the newly diagnosed MM patients with regards to tumour aggressiveness or drug sensitivity. Additionally, we transferred all the Figure panels related to serum M-protein levels to the Supplementary Figure 1 and adjusted the relevant text in all sections of the manuscript to eliminate potential confusions.

The multivariate analysis of ilgL and responses to therapy (page 5, line 115-117): this appears to be based on 24 patients, 12 each in the intracellular Ig Low/high groups. It is not feasible to carry out a meaningful MV analysis with such low patient numbers, as every trial statistician will be able to confirm. Moreover, the CI is sizeable, and only three factors (why these) went into the MV analysis. The data are therefore not convincing. The authors should carry out a proper MV analysis, consider removing the clinical correlation attempts, or find another independent confirmation of their results.

To further strengthen our conclusions about the potential prognostic value of ilgL, we extended the cohort of newly diagnosed MM patients from 24 to 86 patients, where the ilgL concentration was quantified by ELISA (Fig. 1b). From those, 53 patients reached PFS and were included in the correlation and MV analysis that was extended with two additional factors (Fig 1a and Supplementary Table 4). We selected gender/age/ISS stage and LDH in addition to ilgL levels as these are commonly used stratification parameters for MM patients.

Additionally, we performed cubic splines analysis (Fig. 1c) that further supports the prognostic impact of ilgL and validates the selected cut-off value for low and high ilgL in the survival analysis.

Complementary, we now include an independent approach - flow cytometry based on a highly standardized and clinically accepted EuroFlow protocol - to analyse the IgL content in additional 106 newly diagnosed MM patients (Supplementary Fig. 1e).

We believe that these findings based on 1) significantly expanded cohort, 2) additional statistical tests and 3) independent IgL quantification approach support our suggestions of the potential prognostic value of IgL in newly diagnosed MM patients.

Page 7, line 202-203, around IgL knockdown in OTUD1 oe cells (Fig 3c): the authors state that “importantly, the observed changes...directly correlated with the ER stress status”. Where do they show ‘ER stress status’? This is unclear to me. If they mean eIF2a phosphorylation, they need to be clear. eIF2a phosphorylation is also NOT a ‘global ER stress marker’ as the authors claim in their response to my comment 13, where I requested the analysis of a set of ER stress indicators. eIF2a phosphorylation occurs in response to an array of stresses and is carried out by four EIF2AKs. Unless the authors show increased expression on mRNA and/or protein level of adequate ER stress indicators such as chaperones HSPA5/P58IPK, and show PERK (EIF2AK3) phosphorylation, they cannot make any claims of ‘ER stress’. This is a major point that should be addressed. In this context, the authors also need to explore the status of GCN2 (EIF2AK4) and potentially HRI. Both PERK and GCN2, and to some extent HRI, can also be functionally explored with highly selective inhibitors. In particular, GCN2 has recently been identified as a myeloma vulnerability in relation to proteasome inhibition and MYC signalling (PMID: 33883278; PMID: 34732728). Given that eIF2a phosphorylation is such an important marker here, it is critical the authors conduct some experiments that define if GCN2 or PERK are responsible for eIF2a phosphorylation in their experiments. Does PERK or GCN2 inhibition affect any of the downstream effects of eIF2a signalling? One easy and sensitive way is to look for effects on DDIT3 and HSPA5 expression by qRT-PCR/Western. Can the authors also show total eIF2a as a loading control please as it may be cleaved in stressed cells. GCN2 can also be activated by stalling ribosomes, which could be linked to deregulated translation control, so it is worth at least a few experiments as outlined above and a discussion of the roles of GCN2 in MM (PMID: 33883278; PMID: 34732728).

We apologise for confusing the Reviewer regarding our statements about the ER stress status in MM cell lines with altered expression of OTUD1. In the revised version of the manuscript we provide new sets of experiments including analysis of additional ER stress pathways. Specifically, we show ATF6 cleavage (Fig. 3a) and XBP1 splicing (Fig. 3c, d) in two MM cell lines with OTUD1 oe. We believe the presented changes in two another (PERK-independent) ER stress pathways sufficiently confirm upregulation of unfolded protein response (UPR) in MM cells with OTUD1 oe. Therefore, we did not further investigate status of GCN2. Moreover, we now show that OTUD1 has a positive effect on translation elongation (Supplementary Fig. 2e) which further eliminates the possibility of ribosome stalling-induced activation of GCN2.

Response to comment 17 – individual data points: there remains an inexplicably large number of figures in which the authors chose not to show individual data points, when it is clear that they can and should be shown – which is essentially every bar chart. Also, can the authors please confirm in every relevant figure that results are from biologically independent replicates (as opposed to technical replicates) – this is particularly relevant for Fig 2f/g, Fig 3g/h, Fig 6i, Fig 7c. Also, have results shown in Fig. 4g and 5k been repeated?

We apologise for omitting the individual data points from several Figure panels. All the relevant bar graphs in the resubmitted manuscript are shown with the individual data points. Additionally, we completed specifications regarding the character of experimental replicates in the respective Figure legends. Repeats of Fig. 4g, Fig. 5k are included in the Source data file.

Sensitisation to proteasome inhibition: I continue to maintain that short high-dose pulses best reflect PI exposure in vivo in patients, but as there is not agreed consensus for experimental approaches this cannot be resolved objectively, and I am therefore willing to accept their response. However, I urge them to rethink this approach in future work. The fact that others have used low concentrations for longer periods does not make a flawed approach right.

Thank you for your highly relevant suggestion. In our future work, we will re-design the experimental set up in order to apply shorter pulses of high-dose proteasome inhibitors and possibly even compare these approaches site by site.

In this context, another issue is mechanistically troubling, which is the observation that expression of proteasome subunits did not change in OTUD1 oe cells (Suppl Fig 3b-d). If there was indeed a higher load on the proteasome, and ER stress with a UPR, one of the very first responses would be the transcriptional upregulation of proteasomes. How can the authors explain this discrepancy?

We thank the Reviewer for highlighting this interesting phenomenon. Though proteasome genes are generally considered as primary targets of the UPR pathway, we were repeatedly not able to detect any significant changes in the expression of the selected proteasome subunits. We cannot exclude the possibility that expression of certain proteasome subunits (those not included in our list) might be altered in OTUD1 oe cells or that OTUD1 has a yet undescribed role in transcription repression independently of its role in ER stress. Similarly, to our findings, there are multiple evidences indicating increased ER stress does not always translate to enhanced expression of proteasome both in myeloma and other cell types (e.g. PMID: 21252911, PMID: 16103128, PMID: 20651984).

Reviewer #2 (Remarks to the Author):

The authors have addressed most of the issues raised. As a result, several passages have been clarified, and the manuscript grew substantially. However, few concerns have not been understood and still need to be addressed. But more importantly, the story still appears confusing, with its central issue, the link between OTUD1 and immunoglobulin production, being convincingly relevant, but mechanistically unresolved.

In general, the manuscript proves that OTUD1 controls secretory homeostasis and proteasome inhibitor (PI) sensitivity in myeloma cells through a mechanism mediated by the ER-resident peroxiredoxin, PRDX4. The authors also document an anti-proliferative effect of OTUD1, consistent with its proposed oncosuppressive role in other cancers, but this function appears separated from its myeloma-specific role in secretory homeostasis. The paper aims to understand the latter; however, despite remarkable experimental efforts reported in the revised version, the story is still unclear. In a nutshell, OTUD1 increases intracellular Ig in myeloma cells, suggesting that it causes intracellular retention of Ig molecules, perhaps owing to defective Ig maturation along the secretory pathway, but this is in contrast with data that suggest that OTUD1 does not affect Ig mRNA expression, translation, and Ig secretion. The careful reader is left with the doubt as to whether the effect of OTUD1 on Ig folding should be investigated more carefully.

We thank the Reviewer for careful evaluation of our revised manuscript especially in regards of the mechanisms of OTUD1-PRDX4 axis in the Ig production pathway. As the Reviewer points, our work has excluded the involvement of OTUD1 on Ig transcription and secretion. The proteomic analysis of OTUD1 interactors revealed high number of translation and ribosome related proteins (Supplementary Table 7) suggesting OTUD1 might regulate translation of Ig. However, investigation of puromycin incorporation into newly synthesized proteins by the SunSET assay (Supplementary Fig. 2d) revealed no effect of OTUD1.

Based on this result, we incorrectly stated that OTUD1 does not alter Ig translation. The more *precise conclusion should be "OTUD1 does not affect translation initiation" as the SunSET assay does not allow to precisely distinguish separate stages of translation (initiation vs elongation)*. Therefore, we further investigated effect of OTUD1 and PRDX4 on translation elongation. The new data indicate overexpression of OTUD1 (Supplementary Fig. 2e) and PRDX4 (Supplementary Fig. 6a) enhances translation elongation / ribosome processivity. Moreover, the interdomain regions in Ig molecules are likely the sites of ribosome pausing to allow proper folding of Ig domain in ER. Translation pauses would eventually lead to ribosome stalling with suppressive effect on protein synthesis. In support of this hypothesis, it was recently shown that OTUD1 is able to revert stalled ribosomes (PMDI: 32011234).

Collectively, our data indicate the involvement of OTUD1-PRDX4 in the regulation of translation elongation / ribosome processivity might be the possible source of increased Ig production in OTUD1 oe cells. In addition to providing new data related to the role of OTUD1 and PRDX4 in translation elongation, we have also expanded discussion and other relevant sections of the manuscript to provide readers with potential new directions.

The authors also document a specific role of OTUD1 mediating deubiquitination of PRDX4, leading to its protection from ERAD and stabilization, and prove that PRDX4 is causally involved in the mechanism whereby OTUD1 increases intracellular Ig. They conclude that OTUD1 augments Ig production by stabilizing PRDX4, leading to UPS overload and enhanced PI sensitivity. The involvement of PRDX4 is convincing, but its effect on Ig retention and ERAD overload is counterintuitive and unexplained.

Our data show OTUD1 selectively increases stability of PRDX4 while leaving PDI (Fig. 5m) and possibly other components of the ER redox cycle intact. Because PRDX4 is critically needed for re-oxidation of PDI and reduction of molecular peroxide, overrepresentation of only a single component of the ER redox cycle - PRDX4 - might lead to the saturation of the ER folding capacity. This is further potentiated by the positive effect of OTUD1 and PRDX4 on translation processivity. A consequence of a full engagement of the folding apparatus would ultimately be elevation of both folded and misfolded Ig without any apparent changes in Ig transcription, translation initiation and secretion. To clearly delineate this possibility, we extended and clarified our statements and conclusions in the results section and particularly in discussion of the manuscript.

Although a novel role of an OTUD1-PRDX4 axis in PI sensitivity is a relevant and worth reporting finding, since the title and abstract (and entire manuscript) claim that “OTUD1 mediates Ig production”, more conclusive and univocal mechanistic dissection, and thus additional, focused experimental work, in addition to very careful critical discussion, is expected.

In the revised manuscript we now provide a set of new experimental analysis that reveal involvement of OTUD1 and PRDX4 in regulation of translation elongation. In our view, it clarifies the stage and mechanism of how OTUD1 and PRDX4 mediate Ig production as specified above. Additionally, we have added new relevant references and extended all parts of the manuscript especially discussion.

Additional compulsory issues are detailed as follows.

Specific points.

1. Page 7: the title (“OTUD1 modulates IgL folding”) is misleading, as Ig folding has not been directly investigated in this work. Please rephrase.

We agree with the reviewer's opinion and changed the title accordingly.

2. Figure 5: stabilization of PRDX4 by OTUD1 is key in the proposed role played by OTUD 1 in secretory proteostasis. To convincingly prove it, the Authors performed CHX assays, as requested, but the evidence generated completely relies on exogenously expressed proteins. A CHX experiment on endogenous PRDX4 is not only needed, but perfectly feasible in the Authors' hands, as they did measure endogenous PRDX4 abundance upon OTUD1 overexpression (Fig.5i). Essentially, they should simply add CHX treatment to the experiment shown in Fig.5i.

We performed analysis of endogenous PRDX4 in several MM cell lines treated with CHX already for the revised manuscript. However, application of CHX treatment did not lead to any drop in the endogenous PRDX4 protein levels in the time frame and/or under CHX concentrations that did not significantly affect cell viability. Therefore, due to this technical limitation, we believe the use of PRDX4 oe in the CHX assay together with additional data provided in Fig. 5 and Supplementary Fig. 5 are sufficient to claim that OTUD1 regulates PRDX4 stability.

3. Few requests were misinterpreted and not addressed; they should be dealt with.

- In Figure 1, I requested that the results of the CRISPR screening for candidate essential DUBs be displayed as a supplemental item. The authors misinterpreted my request, as they thought to address it by moving a scheme illustrating the strategy (a panel in original Figure 1) to supplemental Figure 1. The scheme was perfectly fine in its original location. Rather, I had requested to "display the results of the screening", i.e., the list of the identified DUBs, e.g. as a supplemental table.

Thank you for clarifying your suggestion. We now provide new Supplementary Table 5 that contains list of DUBs significantly affecting Ig production (based on CRISPR screen data) and MM survival.

- The next request was to "show average quantifications of repeated independent experiments" in protein biochemistry experiments. The Authors respond that they provided original data. That is a laudable action, but again, my request was different. Please do what requested: in all protein biochemistry experiments, in addition to showing representative immunoblots, average quantifications must be provided of the independent experiments run, and the number of independent experiments conducted specified in the legends.

We apology for misunderstanding your request. In the revised version of manuscript, we provide densitometry-based quantification of our biochemistry experiments. The quantification data are presented in the Source Data file together with original uncropped images.

4. Minor issues:

- Abstract, "the combinatory approach": is unclear.

We change the wording to „the CRISPR-based screen and transcriptional analysis“ which precisely reflects the type of analysis.

- Page 10, 10th last line: OTUD1 failed to immunoprecipitate the PRDX4 mutant – not the other way round.

Thank you for spotting this incorrect statement. We changed the text accordingly.

- Page 10, 2nd last line: Fig 5e is in fact 5f.

Thank you for spotting this typo. We changed the Figure description in the text accordingly.

REVIEWERS' COMMENTS

Reviewer #1 (Remarks to the Author):

The authors have addressed relevant points adequately, and the manuscript is acceptable for publication and evaluation by the wider scientific community

Reviewer #2 (Remarks to the Author):

As requested, the new revised manuscript by Vdovin and colleagues contains additional data that contribute to clarify the mechanism linking OTUD1 to Ig synthesis and protein homeostasis in myeloma cells, a mechanism which in the previous revision was still elusive. In particular, as suggested by the authors themselves in their rebuttal, key are the new data defining a previously missed effect exerted by an OTUD1-PRDX4 axis on protein translation (specifically, on translational elongation / ribosome processivity). The data appear critical to the new manuscript. However, these data are presented at distant points of the manuscript and displayed as supplemental items (supplemental figures 2 and 6 and data in supplemental table 7). All data needed to document the new function of OTUD1-PRDX4 on translational elongation should be clearly presented within the same paragraph, with the due focus (including the "high number of translation and ribosome related proteins" among OTUD 1 interactors, which are only commented upon in the response to reviewers), and displayed as main items. Since, as stated by the authors in their rebuttal, these data appear critical to fully understand the otherwise cryptic mechanism underlying OTUD1-driven proteostatic control in MM cells, they should also be shown with the proper quantification of a sound number of independent experiments and proper statistical analyses, which now appear missing (as a general comment, supplementary figures should be as rigorous and conclusive as the main ones).

As a result of the significant experimental efforts made, while the overall content improved critically, the conceptual flow is still rather difficult to follow, even for a reviewer that went through all rounds of revision. My personal view is that additional efforts should be made by the authors to re-think the conceptual flow of the mechanistic part of the manuscript (from figure 2 on) and simplify its structure following the key concepts. This might require moving selected, less central panels from fundamental figures to supplemental ones.

Additional specific issues:

- Issue no. 2. Fig.5 and S5: the data shown suggest, but do not conclusively demonstrate that OTUD1 regulates PRDX4 stability, as it was not directly investigated. Rather, they suggest this possibility in view of the effect of OTUD1 overexpression or silencing on endogenous PRDX4, not accounted for by a similar effect on its transcript. The CHX experiment adopting exogenously expressed proteins (HA-OTUD1 blunting CHX-induced reduction of FLAG-PRDX4) is suggestive, but the authors' statement (in their rebuttal) that endogenous PRDX4 was insensitive to CHX in the same experimental timeframe raises perplexity as it suggests that overexpressed, FLAG-tagged PRDX4 has a different half-life and thus cannot be adopted as a faithful surrogate of the endogenous counterpart. Thus, in the absence of formal experiments exploring PRDX4 stability (pulse-chase radiometabolic assays monitoring the endogenous, immunoprecipitated protein), at lines 268-269 the authors should less squarely conclude "that OTUD 1 deubiquitinates PRDX4 and regulates its abundance, possibly by protecting it from degradation". Similarly, the end of the title of the next paragraph (line 288) should read "through PRDX4" and not "by stabilizing PRDX4".

-Issue no. 3. As already requested, all protein biochemistry data panels and related legends should explicitly indicate if that specific experiment is representative, state the number of independent experiments performed, and display proper quantifications and statistical analyses.

- Minor issue: please check for typos and errors (ex: "raise" for "rise" at line 306).

RESPONSE TO REVIEWERS' COMMENTS

Reviewer #1 (Remarks to the Author):

The authors have addressed relevant points adequately, and the manuscript is acceptable for publication and evaluation by the wider scientific community.

We are grateful the Review for very inspiring comments and suggestions that led to significant improvement of the manuscript.

Reviewer #2 (Remarks to the Author):

As requested, the new revised manuscript by Vdovin and colleagues contains additional data that contribute to clarify the mechanism linking OTUD1 to Ig synthesis and protein homeostasis in myeloma cells, a mechanism which in the previous revision was still elusive.

In particular, as suggested by the authors themselves in their rebuttal, key are the new data defining a previously missed effect exerted by an OTUD1-PRDX4 axis on protein translation (specifically, on translational elongation / ribosome processivity). The data appear critical to the new manuscript. However, these data are presented at distant points of the manuscript and displayed as supplemental items (supplemental figures 2 and 6 and data in supplemental table 7). All data needed to document the new function of OTUD1-PRDX4 on translational elongation should be clearly presented within the same paragraph, with the due focus (including the "high number of translation and ribosome related proteins" among OTUD 1 interactors, which are only commented upon in the response to reviewers), and displayed as main items. Since, as stated by the authors in their rebuttal, these data appear critical to fully understand the otherwise cryptic mechanism underlying OTUD1-driven proteostatic control in MM cells, they should also be shown with the proper quantification of a sound number of independent experiments and proper statistical analyses, which now appear missing (as a general comment, supplementary figures should be as rigorous and conclusive as the main ones).

We thank the reviewer for pointing the translation elongation data as the critical part of the manuscript especially in regards to the OTUD1-PRDX4 driven mechanism of Ig synthesis. As suggested by the reviewer, we have restructured the text by combining the paragraphs and results related to translation control in the section referring to the main Fig 6. Additionally, we clearly specify the possible involvement of OTUD1 in translation processivity in Discussion, rephrase all relevant parts of the text and highlight association of OTUD1 with ribosomal proteins both in the main text and in Supplementary Data 2 file (previously Supplementary Table 7).

As a result of the significant experimental efforts made, while the overall content improved critically, the conceptual flow is still rather difficult to follow, even for a reviewer that went through all rounds of revision. My personal view is that additional efforts should be made by the authors to re-think the conceptual flow of the mechanistic part of the manuscript

(from figure 2 on) and simplify its structure following the key concepts. This might require moving selected, less central panels from fundamental figures to supplemental ones.

In the revisited version, we highlight the effects of OTUD1 and PRDX4 on translation elongation in the main Fig 6 and related text. Specifically, we have moved the respective data (previously in Supplementary Fig 2e and Supplementary Fig 6a) to main Fig 6. Accordingly, we have moved less critical data from Fig 6 to Supplementary Fig 6a, b.

Additional specific issues:

- Issue no. 2. Fig.5 and S5: the data shown suggest, but do not conclusively demonstrate that OTUD1 regulates PRDX4 stability, as it was not directly investigated. Rather, they suggest this possibility in view of the effect of OTUD1 overexpression or silencing on endogenous PRDX4, not accounted for by a similar effect on its transcript. The CHX experiment adopting exogenously expressed proteins (HA-OTUD1 blunting CHX-induced reduction of FLAG-PRDX4) is suggestive, but the authors' statement (in their rebuttal) that endogenous PRDX4 was insensitive to CHX in the same experimental timeframe raises perplexity as it suggests that overexpressed, FLAG-tagged PRDX4 has a different half-life and thus cannot be adopted as a faithful surrogate of the endogenous counterpart. Thus, in the absence of formal experiments exploring PRDX4 stability (pulse-chase radiometabolic assays monitoring the endogenous, immunoprecipitated protein), at lines 268-269 the authors should less squarely conclude "that OTUD 1 deubiquitinates PRDX4 and regulates its abundance, possibly by protecting it from degradation". Similarly, the end of the title of the next paragraph (line 288) should read "through PRDX4" and not "by stabilizing PRDX4".

We agree with the Reviewer that our data regarding the effect of OTUD1 on endogenous PRDX4 protein levels are rather suggestive than definitive. Therefore, we have adequately adjusted our claims and conclusions through the text as the reviewer recommends.

-Issue no. 3. As already requested, all protein biochemistry data panels and related legends should explicitly indicate if that specific experiment is representative, state the number of independent experiments performed, and display proper quantifications and statistical analyses.

We are providing information about number of repeats of each experiment either in the figure legend or in the Statistic and Reproducibility section in the manuscript.

- Minor issue: please check for typos and errors (ex: "raise" for "rise" at line 306).

We thank the reviewer for spotting the spelling issues. The revisited manuscript was proof-read and corrected by a native english speaker.